# Development of emotional labor ability scale for kindergarten teachers

Juan Hong[1,2]*, Ri-Yu Pan[3], Yan-Jin Liu[4]

1 College of Educational Science, Huaihua University, Huaihua, Hunan, China, 2 wulingshan K-12 Educational Research Center at Huaihua University, Huaihua, Hunan, China, 3 School of Teacher Education, Anqing Normal University, Anqing, Anhui, China, 4 International Education Development Research Center, Chengdu Normal University, Chengdu, Sichuan, China

* 3066463476@qq.com

## Abstract

Kindergarten teachers' lack of emotional labor ability can lead to emotional exhaustion, burnout, identity alienation, and low job satisfaction. Systematic training is now an academic consensus. However, a suitable assessment instrument is lacking. This study aims to develop an emotional labor ability scale for kindergarten teachers and test its reliability and validity in China. First, based on the emotional labor working mechanism model, the dimensions of teachers' emotional labor ability were analyzed. Second, Behavioral Event Interview were conducted with 21 teachers to construct primary and secondary competency elements. Two rounds of expert consultation involving 5 experts revised these elements, resulting in an initial questionnaire. Finally, based on survey data from 818 teachers, various analyses (item, exploratory factor, confirmatory factor, reliability, convergence validity, and discriminant validity) were conducted. The scale has 33 items across 5 dimensions: emotional intelligence, internalizing emotional labor rules, coordination in emotional labor, reflection after emotional labor, and applying emotional labor strategies. These dimensions explain 71.757% of the variance. Confirmatory factor analysis shows good model fit ($\chi2/df = 2.774$, RMR = 0.016, GFI = 0.918, IFI = 0.969, TLI = 0.965, CFI = 0.969, RMSEA = 0.047). Internal consistency reliability coefficients range from 0.902 to 0.953. The scale is psychometrically sound and can effectively evaluate kindergarten teachers' emotional labor ability.

## 1 Introduction

Kindergarten teachers are high intensity emotional laborers. Existing research shows that emotional labor not only affects kindergarten teachers on their mental health [1], self-esteem [2] and teaching effectiveness [3], but also has significant predictive effect on the social and emotional development of children [4] as well as the quality of teacher-child interaction [5,6]. However, in educational practice in China, the

**Data availability statement:** All relevant data are within the paper and its Supporting Information files.

**Funding:** This work is supported by the Special scientific research project for on-the-job doctoral students of Chengdu Normal University in 2020" Research on the framework of pedagogy knowledge system under the background of artificial intelligence (ZZBS2020-10)". The funders had no role in study design, data collection and analysis, decision to publish, or preparation of the manuscript.

**Competing interests:** The authors declare that this research was conducted in the absence of any commercial or financial relationships that could be construed as potential conflicts of interest.

emotional labor ability of kindergarten teachers has not been given due consideration in the pre-vocational education, recruitment and post-service certification of kindergarten teachers. As a result, some newly-appointed kindergarten teachers have failed to adapt to the high intensity in emotional labor, which leads to their emotional exhaustion [7], job burnout [8], low job satisfaction [9], and reduced occupational well-being [10]. The fundamental solution to the above problems, suggested by this study, is to improve the emotional labor ability of kindergarten teachers. At present, it has become the academic consensus to carry out systematic training of kindergarten teachers' Emotional labor ability [11].

In the early stages of research on emotional labor ability, existing studies have equated the concepts of "emotional intelligence" and "emotional labor ability" based on the interpersonal interaction characteristics of emotional labor [12]. It is believed that emotional labor ability mainly includes emotional perception ability, emotional understanding ability, emotional management ability, and emotional application ability [13–15]. In terms of the teacher group, some researchers have pointed out that teachers' emotional labor ability includes three components: emotional perception ability, emotional regulation ability, and emotional expression ability [16]. Jennings proposed that emotional skills that teachers need to learn include emotional skills, mindfulness, and care and compassion [17]. With the advancement of research on emotional labor ability, researchers have begun to focus on the specific work situation of workers to explore their emotional labor ability. They believe that emotional labor ability should be manifested as the ability of individuals to use various emotional labor strategies such as deep acting and natural expression based on specific educational situations [18–20]. As for the group of kindergarten teachers, the existing research has focused on the concept [21], importance [22] and promotion strategy [23] of kindergarten teachers' Emotional labor ability by summarizing experience and elaborating views. This provides a useful reference for further exploration on the structural dimension of the emotional labor ability of kindergarten teachers. Based on existing research results, this study believes that the emotional labor ability of kindergarten teachers refers to the ability of kindergarten teachers to actively regulate their internal emotional experiences and external emotional expressions according to the professional requirements of kindergarten teachers in the process of education and teaching, in order to complete educational and teaching tasks.

The existing research results have provided useful references for us to understand the concept of emotional labor ability of kindergarten teachers. However, we need to recognize that the existing research mainly adopts speculative research methods to empirically explore the constituent elements of emotional labor ability of kindergarten teachers, and has not yet adopted empirical research methods to systematically verify and demonstrate the structural dimensions of emotional labor ability of kindergarten teachers. Based on this, there is still significant room for exploration in terms of theoretical appropriateness and practical explanatory power of existing research conclusions on the emotional labor ability of kindergarten teachers. This research gap provides an important opportunity for theoretical innovation and methodological breakthroughs in this study. In view of this, based on the relevant theories on

emotional labor and the practices of emotional labor of kindergarten teachers in China, this study explores the structural dimensions of the emotional labor ability of kindergarten teachers, then further clarifies the logical relationship among its internal factors, and finally develops an emotional labor ability scale for kindergarten teachers that meets the requirements of psychometrics so as to better evaluate and intervene the future training on the emotional labor ability of kindergarten teachers.

## 2. Methods

### 2.1 Research design

This study follows the basic principles of scale development [24]. Firstly, with the theoretical basis of the emotional labor ability of kindergarten teachers being excavated by literature research, the constituent elements of the emotional labor ability of kindergarten teachers are analyzed in this study; Secondly, adhering to the principle of combining theory with practice, behavioral event interviews (BEI) is conducted among in-service kindergarten teachers, and then, by combing the real and specific emotional labor events, this study constructs the primary dimension elements and secondary competency elements of the emotional labor ability of kindergarten teachers; On this basis, two rounds of expert consultation are carried out to revise the dimension elements and competency elements of the emotional labor ability of kindergarten teachers, which results in the Initial Questionnaire of the Emotional Labor Ability of Kindergarten Teachers; Finally, a questionnaire survey are conducted for kindergarten teachers to analyze the reliability and validity of the data obtained, so as to build a scientific and reasonable structure model for the emotional labor ability of kindergarten teachers which results in the formal "Emotional Labor Ability Scale for Kindergarten Teachers".

It should be noted that this study was completed between October 25, 2023 and December 26, 2023. During the process of behavioral event interviews, expert consultations, and questionnaire surveys, all participants provided written informed consent forms. All procedures performed in studies involving human participants were in accordance with the ethical standards of the institutional and/or national research committee and with the 1964 Helsinki Declaration and its later amendments or comparable ethical standards. The study was approved by the Academic Ethics Review Committee of Huaihua University. The ethical approval number is 2023255.

### 2.2 Development and administration of scale

**2.2.1 Theoretical framework.** According to the working mechanism model of emotional labor proposed by Grandey [25], there are three stages in the working mechanism of emotional labor: situational cues, emotional regulation and long-term results (see S1 Fig). In the stage of situational cues, individuals should start by using emotional intelligence to understand the interaction expectations of their work objects and the nature of emotional events, and then they need to have a correct understanding and deep recognition of the emotional labor rule to ensure that the following emotional expression behavior meets the organizational requirements. In the emotional regulation stage, individuals use emotional labor strategies to regulate their external emotional expression behavior and internal emotional feelings under the guidance of the emotional labor rules so that their subjective emotional experience can be transformed into the social action force with educational responsibilities. After emotional labor, individuals need to reflect on both the individual results and organizational results of their emotional labor, including on whether their emotional expression behavior conforms to professional behavior norms or whether it helps to achieve their purposes so as to help promote the emotional labor level of themselves. It can be inferred that during emotional labor, individuals need to be equipped with emotional intelligence, the ability to internalize emotional labor rules, the ability to use emotional labor strategies, and the ability to reflect on their emotional labor.

**2.2.2 Source of initial items.** To explore the special qualities of emotional labor ability of kindergarten teachers, By target sampling, this study conducted BEI for twenty-one in-service kindergarten teachers. The interviewees were from two different types of kindergartens in Hunan Province of China, namely public and private kindergartens, with a working

experience of 3–7 years. Among them, there were 18 female teachers and 3 male teacher. The two interview questions were: What are the emotional labor events that you have done successfully in the past year? What do you think are the components of the emotional labor ability of kindergarten teachers? At the end of the interview, the interviewees were asked to provide their views on revising the dimensions of the emotional labor ability of kindergarten teachers extracted from previous literature research.

Based on their own experience, one of the respondents pointed out, "*I think the ability to lead a class or implement a course can be seen as a guarantee. Because if my teaching ability is relatively weak, an ongoing course will be easily interrupted when emotional events occur. At this time, the emotional state of my own will be affected as well, and this will make the handling of emotional events even harder*(20220913YCY)." Thus it reflects a fact that emotional labor, and teaching activities of kindergartens integrate with each other, which requires a certain coordination ability of kindergarten teachers in emotional labor. In addition, another interviewee emphasized that, "*The guidance of professional codes of conduct, which I believe is the most important, is needed when dealing with emotional events. Otherwise, kindergarten teachers will be likely to lose control when dealing with emotional events, some of which became child abuse events in the past*(20220925DJ)." This view fully explained why the ability of internalizing emotional labor rules plays such an important role as a value guidance among the dimensions of the emotional labor ability of kindergarten teachers. By analyzing and summarizing all interview contents, it is assumed that the emotional labor ability of kindergarten teachers includes five dimensional elements, namely emotional intelligence, the ability of internalizing emotional labor rules, the application ability to emotional labor strategies, the coordination ability in emotional labor, and the reflective ability after emotional labor.

**2.2.3 Delphi method.** Through literature research and BEI analysis, the dimensional elements and competency elements of the emotional labor ability of kindergarten teachers were obtained. Then, two rounds of expert consultation were carried out. The selected expert teachers, closely connected with the management of front-line kindergarten teachers, have been engaged in the teaching of basic theories of preschool education and educational research methods, kindergarten teacher education and other relevant teaching work in colleges or universities for a long time. Their demographic variable characteristics are shown in S1 Table.

The authority coefficient of the five experts is 0.81, indicating a high credibility of the consultation results. The recovery rate of the two rounds of expert consultation questionnaires was 100%, indicating a high level of enthusiasm. In the first round of expert consultation, the average importance of the primary and secondary dimensional elements was: 4.75 points for emotional intelligence, 4.5 points for the ability of internalizing emotional labor rules, 4.75 points for the application ability to emotional labor strategies, 4.75 points for the coordination ability in emotional labor, and 4.5 points for the reflective ability after emotional labor. The average importance of each dimensional element was over 4 points. According to the results of the first round of expert consultation, the dimensional elements and competency elements of the emotional labor ability of kindergarten teachers were modified, and a feedback of the first round was given to all experts in the second round of consultation, and the experts were asked to assign their importance again. In the second round of consultation, the average importance of each dimensional element was: 5 points for emotional intelligence, 4.8 points for the ability of internalizing emotional labor rules, 5 points for the application ability to emotional labor strategies, 5 points for the coordination ability in emotional labor, and 5 points for the reflective ability after emotional labor. The average importance of the above five dimensional elements and their secondary competency elements exceeded 4.5 points, with a full score ratio of 80% and a coefficient of variation of less than 20%, indicating a high consistency of expert opinions. This showed that the design of each dimensional elements and their competency elements for the emotional labor ability of kindergarten teachers is comparatively scientific, and the consistency of experts is high. As a result, an initial model of the emotional labor ability of kindergarten teachers, consisting of 5 basic elements and 16 competency elements was built (see S2 Fig), based on which the Initial Questionnaire of the Emotional Labor Ability of Kindergarten Teachers was compiled.

**2.2.4 Questionnaire survey.** The questionnaire comprises five dimensional elements and 37 items, with each question item adopting the Likert five-point scoring method to facilitate quantitative analysis. Previous studies have

shown that survey data with large sample sizes, diverse sample regions, and random sampling methods can improve sample representativeness and reliability, making research results more likely to approach the true characteristics of the population [26,27]. Therefore, based on the researcher's existing research cooperation networks in Xinjiang Uygur Autonomous Region (western region), Hunan Province (central region), and Jiangsu Province (eastern region), we used convenience sampling to determine the sample area. This design ensures that the sample covers the three major economic geographical zones of eastern, central, and western China, thereby enhancing the geographical representativeness of the sample. In the selected area, we distributed questionnaires to in-service kindergarten teachers through random sampling to ensure that each teacher has an equal probability of being included in the study and avoid human selection bias. Ultimately, 818 valid questionnaires were collected. Among them, Jiangsu Province collected 294 valid questionnaires, Hunan Province collected 366 valid questionnaires, and Xinjiang Uygur Autonomous Region collected 158 valid questionnaires. The sample size has covered the eastern, central, and western regions of China. In addition, Previous studies have suggested that the number of valid questionnaires should be at least 10 times the number of items on the scale to be more reliable [28]. The number of valid questionnaires in this study is 22.1 times the total number of questions, which is much larger than the theoretical sample threshold (37 × 10). In summary, this study demonstrates high representativeness and diversity with its large sample size, extensive geographical coverage, and random sampling method.

In the valid questionnaire, there were 50 male kindergarten teachers, accounting for 6.11% of the total number, and 768 female kindergarten teachers, accounting for 93.89% of the total number. After collecting valid data, use SPSS software to randomly split the data into two parts. The first 409 pieces of data were used for exploratory factor analysis (EFA), which aimed to identify the underlying factors or dimensions of the questionnaire. The other 409 pieces of data were reserved for confirmatory factor analysis (CFA), which was conducted to verify the factor structure identified in the EFA and assess the questionnaire's construct validity. Overall, this study employed several analytical methods including item analysis, exploratory factor analysis, confirmatory factor analysis, reliability and Validity analysis. After collecting valid data, use SPSS software to randomly split the data into two parts. The first 409 pieces of data were used for exploratory factor analysis (EFA), which aimed to identify the underlying factors or dimensions of the questionnaire. The other 409 pieces of data were reserved for confirmatory factor analysis (CFA), which was conducted to verify the factor structure identified in the EFA and assess the questionnaire's construct validity. Overall, this study employed several analytical methods including item analysis, exploratory factor analysis, confirmatory factor analysis, reliability and Validity analysis.

## 3. Results

### 3.1 Item analysis

The total score of each subject on the scale was calculated and then ranked in ascending order. The top 27% were the low-score groups while the bottom 27% were the high-score groups. Independent t-tests were conducted on the scores of the high and low groups for the 37 items, and subsequently, the Pearson correlation between each item and the total score was calculated. The results, as can be seen from S2 Table, showed that the values of t range from −16.07 to −7.07, and the p-values are all less than 0.001. Correlation analysis showed that the correlation coefficient between each item and the total score is between 0.57 and 0.81, all exceeding 0.4, and the p-value is less than 0.001. The internal consistency test showed that the Cronbach's alpha coefficient of the whole scale is 0.98. Additionally, the Cronbach's alpha coefficient when deleting a certain item was calculated,too. And the results showed that deleting any item would result in a decrease in reliability. Therefore, the item analysis of this study did not delete any item.

### 3.2 Exploratory factor analysis

The Initial Questionnaire of the Emotional Labor Ability of Kindergarten Teachers was tested by Kaiser-Meyer-Olkin (KMO) and Bartlett's test of sphericity. The results showed that the KMO coefficient was 0.966, the value of $\chi^2$ was 13568.248, the degree of freedom was 666, and $p < 0.001$, indicating that it is advisable to conduct exploratory factor analysis.

Using principal component analysis and maximum variance method to extract common factors, and then deleting items with factor load value less than 0.5, commonality less than 0.4, multiple loads with similar load values, or improper factor classification, a total of "E1, E2, E3, B2" items were deleted, with 33 items remaining. Finally, five common factors were obtained, with a cumulative total variance explained of 71.757%, the resulting item load values and commonality are shown in S3 Table.

### 3.3  Confirmatory factor analysis

Confirmatory factor analysis was conducted using the software of Amos 21.0. According to the modification indices and factor loads, A8, D1, and D2 were deleted and correlations for residuals between certain items were established based on theoretical analysis.

To establish an optimal model, multiple validations were conducted on the model(see S4 Table) and it is found that the $\chi^2$/df, GFI, IFI, TLI, CFI, and RMSEA in one factor model did not reach the recommended values, while all indicators of the five-factor model met the recommendation requirements. This indicated that the five-factor model fits well and is in line with the actual situation. The final model is shown in S3 Fig.

### 3.4  Reliability and validity testing

As can be seen from S5 Table, the reliability analysis showed that the Cronbach's α of the Emotional Labor Ability Scale for Kindergarten Teachers is 0.974, indicating a good reliability of the scale. In addition, for each factor, the coefficients Cronbach's α are all above 0.902, indicating a good reliability of each factor.

Confirmatory factor analysis (CFA) was conducted for five factors and thirty items. As shown in S6 Table, the AVE values of the three factors were all greater than 0.5, and the CR values were all higher than 0.7, which means that the scale had good convergence validity.

According to the analysis of discriminant validity (S7 Table), the square root of AVE square root values for the factors were 0.748, 0.801, 0.876, 0.841 and 0.775, which were greater than the maximum value of absolute correlation coefficient between factors, meaning that the scale had good discriminant validity.

### 3.5  Descriptive statistics

According to the analysis of Descriptive Statistics (S8 Table), The average values of emotional intelligence, the ability of internalizing emotional labor rules, the coordination ability in emotional labor, the reflective ability after emotional labor, and the application ability to emotional labor strategies in each dimension are 3.94, 4.05, 4.00, 3.99, and 3.97, respectively. The mean values of all dimensions are significantly higher than the theoretical median, indicating that the surveyed group of kindergarten teachers generally self evaluate to have high emotional labor ability. The standard deviation of each dimension ranges from 0.54 to 0.61, and the variance ranges from 0.30 to 0.38, indicating a relatively low degree of data dispersion.

## 4.  Discussion

This study aims to develop the Emotional Labor Ability Scale for Kindergarten Teachers in order to provide an effective tool for evaluating the emotional labor ability of kindergarten teachers. Therefore, based on the working mechanism model of emotional labor, this study conducted behavioral event interviews with front-line kindergarten teachers, followed by two rounds of expert consultation. As a result, the dimensional elements and competency elements of the emotional labor ability of kindergarten teachers were built and the Initial Questionnaire of the Emotional Labor Ability of Kindergarten Teachers was complied. After that, questionnaire surveys were conducted among kindergarten teachers, and then item analysis, exploratory factor analysis, confirmatory factor analysis, reliability analysis, convergence validity analysis, and discriminant validity analysis on the valid data obtained were conducted, which proved that the Emotional Labor Ability Scale for

Kindergarten Teachers is highly reliable and valid. The scale has 30 items and 5 dimensions, namely emotional intelligence, the ability of internalizing emotional labor rules, the coordination ability in emotional labor, the application ability to emotional labor strategies, and the reflective ability after emotional labor.

### 4.1 Emotional intelligence: Permeating all stages of the emotional labor of kindergarten teachers

Emotional intelligence refers to the ability of kindergarten teachers to recognize and understand the emotional state of others in interpersonal interaction, and to use this information to guide the whole process of emotional labor. It includes two sub-dimensions, namely the ability of emotional recognition and the ability of emotional understanding, and a total of 8 items. Young children have variable emotions but weaker ability in language expression and they have not yet, been able to accurately express their emotional feelings and interactive demands. Therefore, in the stage of situational cues, kindergarten teachers need to use their emotional intelligence to identify and understand the emotional information conveyed by children, which has a significant impact on the emotional labor level of kindergarten teachers [29]. In the emotional regulation stage, kindergarten teachers with high emotional intelligence can use emotional regulation strategies more flexibly [30]. After emotional labor, emotional intelligence can also alleviate the emotional distress [31] and job burnout [32] caused by kindergarten teachers during their work process. Based on this, emotional intelligence is the psychological cornerstone for kindergarten teachers to carry out emotional labor, and it plays a fundamental, guiding and overall role in all stages of emotional labor of kindergarten teachers. The higher the score in this dimension, the more acutely kindergarten teachers can recognize the emotional information transmitted in the process of emotional labor, and thus correctly understand the meaning of emotional expression and respond in a timely manner.

### 4.2 The ability of internalizing emotional labor rrules: A value guidance for emotional labor

The ability of internalizing emotional labor rules refers to the ability of kindergarten teachers to deeply identify with and consciously practice the code of conduct of emotional labor for the normal operation of emotional labor, including the subconsciousness of emotional labor rules and the automation of emotional labor rules, with a total of 5 items. Research has shown that the emotional labor rule is the bumper for kindergarten teachers in the process of Emotional labor [33]. When kindergarten teachers identify with the rules of emotional labor, they can avoid emotional exhaustion in the process of Emotional labor. Therefore, kindergarten teachers need to consciously follow the corresponding rules in emotional labor. Only when kindergarten teachers internalize the rules of emotional labor in their minds and externalize them in their actions, can they follow their own inclinations without violating the rules in emotional labor. More importantly, the extent to which kindergarten teachers identify with emotional labor rules will directly affect their choices of emotional labor strategies [34,35]. Therefore, it can be concluded that the ability of internalizing emotional labor rules is the value guidance for kindergarten teachers to carry out Emotional labor, and determines and maintains that kindergarten teachers' Emotional labor conforms to the standard. The higher the score in this dimension, the more kindergarten teachers identify with the requirements of the organization for emotional labor, and the more they can protect the interests of the organization as well as their professional image in emotional labor.

### 4.3 The coordination ability in emotional labor: A guarantee for smooth progress of emotional labor

The coordination ability in emotional labor refers to the ability of kindergarten teachers to deal with emotional events in complex educational situations while ensure the smooth progress of all aspects of the kindergarten's daily activities. It mainly includes six sub-dimensions, namely the ability to coordinate collective teaching activities of the kindergarten, the ability to coordinate the morning activities of children, the ability to coordinate the life activities of children, the ability to coordinate the activities in corresponding activity area, and the ability to coordinate the outdoor or after-school activities of children, totaling 6 items.The emotional labor of kindergarten teachers permeates in all aspects of the kindergarten's daily activities, and they are integrated and inseparable. Existing research also suggests that different emotions can become

professionalized based on the job content of the laborer themselves [36]. As far as preschool education is concerned, emotional labor is indeed, in a professional development where it co-exists and coordinates with kindergarten teaching activities. Bringing the coordination ability in emotional labor into the concept of the emotional labor ability of kindergarten teachers also responds to the view of previous studies, that is, emotional labor should no longer be regarded as a means of exploitation [37] but as an indispensable part of professional practical activities in preschool education [38]. The higher the score in this dimension, the more control kindergarten teachers have in emotional ability, and the more like they can strike a perfect balance between high-intensity emotional labor and professional teaching activities.

### 4.4 The application ability to emotional labor strategies: The key to effective operation of emotional labor

The application ability to emotional labor strategies refers to the ability of kindergarten teachers to use certain emotional labor strategies according to the needs of educational situations in the process of emotional labor, including two sub-dimensions, namely the ability to use deep performance strategy and the ability to use natural performance strategy, with a total of 6 items. It is worth noting that emotional labor strategies include the strategies of surface acting, deep acting and natural expression [39]. However, in the process of exploratory factor analysis, the E1, E2, and E3 items under the dimension of "the application ability to surface acting strategy" were excluded due to the occurrence of cross loading. The reason may be that the surface acting strategy belongs to a remedial process of emotional labor [25], that is, When there is a certain gap between the true emotional feelings of kindergarten teachers and the emotional labor rules stipulated by the organization, kindergarten teachers will use surface acting strategies to cope. This means that kindergarten teachers only change their external emotional expression to meet organizational requirements, but their inner selves do not fully agree with the laws of emotional labor, which can lead to emotional disorders, emotional exhaustion, professional burnout, and other situations among kindergarten teachers [40]. Therefore, the use of surface acting strategy cannot be regarded as a symbol of competence in emotional labor. In general, the higher the score of this dimension, the more frequently kindergarten teachers adopt deep acting and natural expression strategies in the process of emotional labor, and the more appropriate emotions they can express in complex and changing educational situations to meet organizational requirements.

### 4.5 The reflective ability after emotional labor: The power source for the circulation of emotional labor

The essence of reflective ability after emotional labor is that kindergarten teachers actively review their own emotional labor process to find their own problems in emotional labor and to find effective ways to solve problems, including three sub-dimensions, namely reflective perseverance, the ability to reflect on relevant theoretical knowledge and the ability to reflect on emotional labor experience, with a total of 8 items. The reflective ability after emotional labor is an important symbol of the subjectivity of kindergarten teachers in emotional labor. It is also a high-level cognitive activity for kindergarten teachers to critically review the process and results of emotional labor. Moreover, reflective ability is an important component of professional growth [41] as well as an inevitable requirement for the development of teachers as a reflective educator [42]. Therefore, improving reflective ability after emotional labor is the internal need of kindergarten teachers in their professional development. The higher the score in this dimension, the more actively kindergarten teachers can participate in the process of emotional labor, the more independent thinking and rational analysis can be carried out in the process of emotional labor so that their emotional labor level can be constantly improved.

This study has certain theoretical significance. First, the five-factor structure of the emotional labor ability of kindergarten teachers constructed in this study is basically consistent with the working mechanism model of emotional labor proposed by Grundi, which helps to demonstrate the explanatory power of the working mechanism model of emotional labor in the field of education. Secondly, previous studies mainly focused on one aspect of emotional labor ability, such as emotional intelligence, application ability to emotional labor strategy, etc. Yet this study conducted a systematic and comprehensive analysis on the constituent elements of the emotional labor ability of kindergarten teachers, and proposed

 

a five-factor model of the emotional labor ability of kindergarten teachers, which can enrich the connotation of the emotional labor ability of kindergarten teachers. Finally, most of the existing studies use a single speculative research method to describe the components of the emotional labor ability of kindergarten teachers. Thus it remains unknown whether the deductive reasoning by personal perceptual experience can withstand data verification, which leads to a relative lack of feasibility and applicability of research conclusions. This study used mixed research methods to verify and analyze the internal structure of the emotional labor ability of kindergarten teachers, and developed the Emotional Labor Ability Scale for Kindergarten Teachers that meets the requirements of psychological measurement, which has certain practicality and can be provided as an effective tool for the development of quantitative research in this field in the future.

This study has certain practical significance, too. The Emotional Labor Ability Scale for Kindergarten Teachers developed in this study can be used by kindergarten managers for the evaluation of the emotional labor ability of applicants, so as to change the situation where kindergarten teacher recruitment only focuses on the teaching ability of kindergarten teachers and ignores their emotional emotional ability. In doing so, job seekers who do not match the kindergarten-teacher career can be eliminated from the start to maximize the personnel and job matching of kindergarten teachers so that the stability of the kindergarten teacher team can be effectively promoted. In addition, this scale can be used as an evaluation tool for the effectiveness of emotional labor ability training for kindergarten teachers. The emotional labor ability of trainees can be measured before and after the training activities. By comparing the scores of kindergarten teachers at different stages of the training activities, the effectiveness of emotional labor ability training can be judged, and the most scientific and effective training methods can be selected to efficiently coordinate post service training resources for kindergarten teachers.

## 5. Limitations and future research

Although the reliability and validity test results of the scale shows that the Emotional Labor Ability Scale for Kindergarten Teachers developed in this study can be used as an effective tool to evaluate the emotional labor ability of kindergarten teachers, there are still some limitations and shortcomings. First, the emotional labor ability of kindergarten teachers is a complex concept involving psychology, pedagogy, sociology and other multi-disciplinary fields. This study divides it into five dimensions: emotional intelligence, the ability of internalizing emotional labor rules, the coordination ability in emotional labor, the application ability to emotional labor strategies, and the reflective ability after emotional labor, which may not cover all the connotations of the concept. Therefore, whether the five-dimension structure proposed in this study is the optimal structure for the emotional labor ability of kindergarten teachers remains to be further verified by future research. Second, although the scale has been formed based on solid theoretical foundations, two rounds of expert opinions, interviews with grassroots kindergarten teachers, and a large amount of questionnaire survey data, research results also indicate that the scale has good psychological measurement characteristics. However, the lack of demographic variable information may lead to limited universality of research results. Therefore, in the future stage of large sample surveys, we will include core demographic indicators such as age, teaching experience, education, kindergarten type, etc. for further in-depth exploration to improve the representativeness and universality of research results. Third, the Emotional Labor Ability Scale for Kindergarten Teachers developed in this study is more suitable for the kindergarten teachers in the Chinese educational context, and its relevance in a cross-cultural context needs further testing. Therefore, more research is needed to determine whether the scale has certain cross-cultural invariance.

## 6. Conclusion

The Emotional Labor Ability Scale for Kindergarten Teachers, developed through rigorous research in this study, comprises five comprehensive dimensional elements. These dimensions include emotional intelligence, the proficiency in internalizing emotional labor rules, the coordination ability essential for managing emotional labor effectively, the practical application ability to implement emotional labor strategies, and the reflective ability that follows emotional labor

experiences. These dimensions are further broken down into 15 distinct competency elements, which are assessed through a total of 30 meticulously crafted items.The scale has undergone a thorough validation process, encompassing item analysis, exploratory factor analysis, confirmatory factor analysis, reliability analysis, convergence validity analysis, and discriminant validity analysis. The results from these analyses consistently demonstrate that the scale possesses robust psychometric properties. It exhibits high reliability, indicating consistent and stable measurements over time. Furthermore, the scale demonstrates strong convergence validity, confirming that it accurately measures the intended constructs related to emotional labor ability in kindergarten teachers. Additionally, the discriminant validity analysis confirms that the scale distinguishes between different constructs, ensuring its specificity and precision.

## Supporting information

**S1 Fig.  Working Mechanism of Emotional Labor.**
(DOCX)

**S2 Fig.  The Initial Model for the Emotional Labor Ability of Kindergarten Teachers.**
(DOCX)

**S3 Fig.  The Final Model of the Emotional Labor Ability of Kindergarten Teachers.**
(DOCX)

**S1 Table.  Demographic Characteristics of the Experts.**
(DOCX)

**S2 Table.  Correlation Coefficients Between Items and Total Score.**
(DOCX)

**S3 Table.  Exploratory Factor Analysis Results of the Questionnaire of the Emotional Labor Ability of Kindergarten Teachers.**
(DOCX)

**S4 Table.  Fitting Indicator of the Model of the Emotional Labor Ability of Teachers.**
(DOCX)

**S5 Table.  Coefficient Cronbach's α.**
(DOCX)

**S6 Table.  Model AVE and CR Indexes Results.**
(DOCX)

**S7 Table.  Pearson Correlation and AVE Square Root Value.**
(DOCX)

**S8 Table.  Descriptive Statistics(N = 818).**
(DOCX)

## Author contributions

**Conceptualization:** Juan Hong.

**Data curation:** Ri-Yu Pan.

**Formal analysis:** Juan Hong.

**Funding acquisition:** Juan Hong.

**Investigation:** Juan Hong.

**Methodology:** Juan Hong.

**Project administration:** Juan Hong.

**Resources:** Juan Hong.

**Software:** Juan Hong.

**Supervision:** Juan Hong, Yan-Jin Liu.

**Validation:** Ri-Yu Pan.

**Visualization:** Juan Hong.

**Writing – original draft:** Juan Hong.

**Writing – review & editing:** Juan Hong, Yan-Jin Liu.

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
