## [Decision Letter · Decision Letter 0]

Dear Dr. Hong,

Thank you for submitting your manuscript to PLOS ONE. After careful consideration, we feel that it has merit but does not fully meet PLOS ONE’s publication criteria as it currently stands. Therefore, we invite you to submit a revised version of the manuscript that addresses the points raised during the review process.

The manuscript has been evaluated by three reviewers, and their comments are available below, and in the attached document.



We look forward to receiving your revised manuscript.

Kind regards,

Steve Zimmerman, PhD

Senior Editor, PLOS One

“This paper is a phased research result of the 2022 special project of the Xinjiang Uygur Autonomous Region Social Science Fund “Research on Education Assistance in Xinjiang form the Perspective of Nurturing Xinjiang with Culture” (Grant No: 22VZX008).”

6. We are unable to open your Supporting Information file [Original data.sav]. Please kindly revise as necessary and re-upload.

Reviewers' comments:

Reviewer's Responses to Questions

**Comments to the Author**

1. Is the manuscript technically sound, and do the data support the conclusions?

Reviewer #1: Partly

Reviewer #2: No

Reviewer #3: Yes

2. Has the statistical analysis been performed appropriately and rigorously?

Reviewer #1: Yes

Reviewer #2: No

Reviewer #3: Yes

3. Have the authors made all data underlying the findings in their manuscript fully available?

Reviewer #1: Yes

Reviewer #2: No

Reviewer #3: Yes

4. Is the manuscript presented in an intelligible fashion and written in standard English?

Reviewer #1: Yes

Reviewer #2: No

Reviewer #3: Yes

Reviewer #1: To the question "Is the manuscript technically sound, and do the data support the conclusions?" I answered "Partly" because I am nor sure whether interviews of 5 kindergarten teachers are a sufficient sample for analysis. At least 10, better 30 interviewees would be better.

I Have several technical issues with the article:

1) Several references (both in-text and in the list) are done not quite adequately: It should be Surname, initials and not full names [1,4,8,10,16, 17, 18,25, 26, 31, 32, 33]. Writing Grandey, & A. or Grandey, & Alice, as well as Beauchamp, & Catherine is incorrect. It should be Grandley, A.; Beauchamp, C.

2) There are some minor spelling mistakes, e.g.: p. 29 the extend to which (should be the extent to which)

3) There are some capitalization issues, e.g.: in the abstract "Methods" should be capitalized, in 2.2.4 questionnaire should be capitalized. Also in chapter 4 subtitles should be fully capitalized, not only the first words.

4) In section 3.1 the sentence "Conduct independent t-tests, ..." is in imperative mood, but it should be in Past Simple test (Independent t-tests were conducted, ..."

Reviewer #2: Thank you for the opportunity to review the manuscript.

The manuscript reports a study aiming to develop a scale titled Emotional Labor Ability Scale for Kindergarten Teachers.

Although the study certainly has its merits, such as adhering to protocols for scale development, it unfortunately contains a fundamental flaw: the lack of a conceptual or theoretical underpinning for the term “emotional labor ability.” The term “emotional labor ability” does not make sense. We can use terms like “emotional ability,” “emotional competence,” “emotional intelligence,” or “emotional skills,” but “emotional labor ability” is not appropriate. This is because emotional labor comprises several distinct constructs (i.e., surface acting, deep acting, and the expression of naturally felt emotions). The performance of these constructs cannot be aggregated into what might be termed an “ability.” For instance, a teacher who performs surface acting more does not necessarily have higher “emotional labor ability,” and the same applies to deep acting and the expression of naturally felt emotions. Throughout the literature on emotional labor, no researcher attempts to measure an “ability” related to emotional labor, despite the existence of numerous scales measuring emotional labor constructs. The absence of a scale measuring “emotional labor ability” is not a research gap but a logical reality; such an “ability” does not exist.

The authors claim that Grandey (2000) serves as the theoretical framework for the study. However, Grandey’s framework does not support the concept of “emotional labor ability.” Grandey’s framework is based on situational cues, emotional labor (deep acting, surface acting), and long-term consequences. In contrast, the “emotional labor ability” scale in this study includes emotional intelligence, the ability to internalize emotional labor rules, coordination in emotional labor, reflective ability after emotional labor, and the application of emotional labor strategies. Clearly, the five constructs of “emotional labor ability” in this study are not supported by existing literature.

I believe that this fundamental flaw alone renders the study illegitimate, regardless of how the manuscript is revised.

Reviewer #3: A good study but requires Proofreading. It's important to address what are written in the comments. Kindly focus on the elaboration for Method and Discussion sections. The Results should be well addressed.

**Do you want your identity to be public for this peer review?** For information about this choice, including consent withdrawal, please see our Privacy Policy

Reviewer #1: No

Reviewer #2: **Yes: ** Qilong Zhang

Reviewer #3: No

---

## [Author Response · Author response to Decision Letter 1]

9 Mar 2025

Report

Reviewer1 09.03.2025

1.Several references (both in-text and in the list) are done not quite adequately: It should be Surname, initials and not full names [1,4,8,10,16, 17, 18,25, 26, 31, 32, 33]. Writing Grandey, & A. or Grandey, & Alice, as well as Beauchamp, & Catherine is incorrect. It should be Grandley, A.; Beauchamp, C.

Thanks for the comment. Based on the comment, the results were included in the references

[1]Byeon, H., Yim, M S.. (2021). Mediating Effect of Resilience on the Relationship between Emotional Intelligence and Emotional Labor. Logos Management Review, 19(2), 97-114. https://doi.org/10.22724/LMR.2021.19.2.97.

[4]Cho,E., SUKYUNG, S.. (2022). Mediating Effect of Autonomy on the Relationship Between Self-leadership and Emotional Labor of Early Childhood Teachers. The Journal of Humanities and Social science, 13(6), 827-842. https://doi.org/10.22143/hss21.13.6.56.

[8]EUN,Y J. (2022). Analyze Factors to Influence the Turnover Intention of Childcare Teachers. The Journal of the Korea Contents Association, 22(11), 660-672. https://doi.org/10.5392/JKCA.2022.22.11.660.

[10]Grandey,A.. (2000). Emotion regulation in the workplace: a new way to conceptualize emotional labor. J Occup Health Psychol, 5(1), 95-110. https://doi.org/10.1037/1076-8998.5.1.95.

[16]Kyung,M Y.,Jang, H J.. (2021). Childcare Teacher’s Emotional Labor Clusters: The Differences of Inter-Personal Stress and Self-Esteem by Clusters. The Society of Convergence Knowledge Transactions, 9(1), 75-83. https://doi.org/10.22716/sckt.2021.9.1.008.

[17]Kwak,E., So-Hyun,J.. (2022). The Effect of Early Childhood Teacher’s Emotional Labor on Child-teacher Interaction: Serial Multiple Mediation of Teaching Efficacy and Burnout. Journal of Children’s Literature and Education, 23(3), 297-312. https://doi.org/10.22154/jcle.23.3.13.

[18]Kyung-chul,K.,Zhang,X Y .., Chunjing, Cai.. (2021). The Effect of Interpersonal Relationships in the Workplace of Kindergarten Teachers in China on Job Burnout: Focusing on the Emotional Intelligence Modulating Effect. International Journal of Early Childhood Education, 27(2), 89-115.

[25]No, S., Yeong-hee, K.. (2021) Effects of Preschool Teacher’s Emotional Labor on Burnout, Job Satisfaction, and Turnover Intention. Life Science Research Forum, 25(1), 69-81. https://doi.org/10.36357/johe.2021.25.1.69.

[26]Nan-Sil, K., Suhyun, K.. (2020). The Effects of Childcare Teachers’ Emotional Labor on Psychological Well-being. The Journal of Humanities and Social science, 11(5), 1659-1674. https://doi.org/10.22143/hss21.11.5.118.

[31]Shin, D., Myoung, U S.. (2020). A Study on the Relations among Emotional Labor Types, Perception of Emotional Display Rules and Emotion Regulation Strategies of Childcare Center Teachers. Korean Journal of Child Education and Care, 20(3),25-40. https://doi.org/10.21213/kjcec.2020.20.3.25.

[32]Seom, H S.. (2022). An Analysis on Kindergarten Teachers’ Emotional Labor. The Journal of EducationKyungin University of Education Educational Research Institute,42(3),137-151. https://doi.org/10.25020/je.2022.42.3.137.

[33]Shim, D H,. (2022) The Main and Interaction Effect of Deep Emotional Labour and Supervisor’s Emotional Support on Teacher-Child Emotional Interaction in Childcare Center Teachers. The Korea Association of Child Care and Education, 132, 55-76. https://doi.org/10.37918/kce.2022.1.132.55.

2.There are some minor spelling mistakes, e.g.: p. 29 the extend to which (should be the extent to which)

Thanks for the comment. Based on the comment, Modified in the paper.

“The extend to which” has been modified to“the extent to which”

3.There are some capitalization issues, e.g.: in the abstract "Methods" should be capitalized, in 2.2.4 questionnaire should be capitalized. Also in chapter 4 subtitles should be fully capitalized, not only the first words.

Thanks for the comment. Thanks for the comment.

I have made adjustments to the expression of this sentence in the text.

4. In section 3.1 the sentence "Conduct independent t-tests, ..." is in imperative mood, but it should be in Past Simple test (Independent t-tests were conducted, ..."

Thanks for the comment.

I have made adjustments to the expression of this sentence in the text.

Independent t-tests were conducted on the scores of the high and low groups for the 37 items, and subsequently, the Pearson correlation between each item and the total score was calculated.

Report

Reviewer 2 09.03.2025

1.The manuscript reports a study aiming to develop a scale titled Emotional Labor Ability Scale for Kindergarten Teachers.

Although the study certainly has its merits, such as adhering to protocols for scale development, it unfortunately contains a fundamental flaw: the lack of a conceptual or theoretical underpinning for the term “emotional labor ability.” The term “emotional labor ability” does not make sense. We can use terms like “emotional ability,” “emotional competence,” “emotional intelligence,” or “emotional skills,” but “emotional labor ability” is not appropriate. This is because emotional labor comprises several distinct constructs (i.e., surface acting, deep acting, and the expression of naturally felt emotions). The performance of these constructs cannot be aggregated into what might be termed an “ability.” For instance, a teacher who performs surface acting more does not necessarily have higher “emotional labor ability,” and the same applies to deep acting and the expression of naturally felt emotions. Throughout the literature on emotional labor, no researcher attempts to measure an “ability” related to emotional labor, despite the existence of numerous scales measuring emotional labor constructs. The absence of a scale measuring “emotional labor ability” is not a research gap but a logical reality; such an “ability” does not exist.

The authors claim that Grandey (2000) serves as the theoretical framework for the study. However, Grandey’s framework does not support the concept of “emotional labor ability.” Grandey’s framework is based on situational cues, emotional labor (deep acting, surface acting), and long-term consequences. In contrast, the “emotional labor ability” scale in this study includes emotional intelligence, the ability to internalize emotional labor rules, coordination in emotional labor, reflective ability after emotional labor, and the application of emotional labor strategies. Clearly, the five constructs of “emotional labor ability” in this study are not supported by existing literature.

I believe that this fundamental flaw alone renders the study illegitimate, regardless of how the manuscript is revised.

Thanks for the comment. Based on the comment, The author makes the following explanation and makes some modifications in the article.

1 The Concept of "Emotional Labor Ability" Exists

Firstly, it must be emphasized that "emotional labor ability" is not a non-existent concept. Rather, the term "emotional labor ability," as an academic concept, has been widely used by both domestic and international academic communities.

For instance, in the medical field, studies have indicated that possessing emotional labor ability is of great significance for establishing harmonious doctor-patient and nurse-patient relationships, as well as for improving patient satisfaction. Having emotional labor ability is of great significance to build a harmonious doctor-patient relationship, nurse-patient relationship, and to improve patients' satisfaction.[[[]Juan Z,et,al.. (2020). Application and prospect of emotional labor in nursing research. Chinese Nursing Management. 20(10):1528-1533.]]Moreover, research has also investigated the impact of narrative dialogue on the emotional labor capacity of male psychiatric nurses with lower levels of experience.Previous studies have explored the impact of narrative dialogue on the emotional labor ability of low qualified male psychiatric nurses.[[[]Lingfang, Z. , Jianying, W. , & Donghong, Z. . (2019). Effect of narrative dialogue on emotional labor ability in male psychiatric nurses with low seniority. Journal of Psychiatry.]]

For the group of kindergarten teachers, scholars have emphasized the importance of conducting training programs aimed at enhancing their emotional labor ability[[[]Cho, E., SUKYUNG, S.. (2022). Mediating Effect of Autonomy on the Relationship Between Self-leadership and Emotional Labor of Early Childhood Teachers.The Journal of Humanities and Social science, 13(6):827-842.]]. Furthermore,A study in South Korea, based on positive psychology, developed an emotional labor coping ability enhancement plan for preschool teachers and applied it to preschool teachers, achieving relatively positive experimental results.[[[]On, E.. (2022). The Development and Effect of Positive Psychology-based Managing Skill Improvement Program for Emotional Labors of Kindergarten Teachers. The Journal of Learner-Centered Curriculum and Instruction. 22(19): 217-231.]]Of particular importance is that enhancing the emotional labor ability of kindergarten teachers has become a fundamental consensus within the academic community both domestically and internationally[[[]No, S k., Yeong-hee, K.. (2021) Effects of Preschool Teacher’s Emotional Labor on Burnout, Job Satisfaction, and Turnover Intention. Life Science Research Forum, 25(1), 69-81. DOI:10.36357/johe.2021.25.1.69.]][[[]Peng, J., He, Y., Deng, J., Zheng, L., Chang, Y., & Liu, X.. (2019). Emotional labor strategies and job burnout in preschool teachers: psychological capital as a mediator and moderator. Journal of neurosurgical sciences(3), 63. DOI:10.3233/WOR-192939.]][[[]EUN, Y J. (2022). Analyze Factors to Influence the Turnover Intention of Childcare Teachers. The Journal of the Korea Contents Association, 22(11), 660-672. DOI:10.5392/JKCA.2022.22.11.660.]][[[]Wei S D., Guan J F., Wang S M., Liang J Q..(2021) The relationship between professional identity, emotional labor, and professional well-being among kindergarten teachers [J]. China Journal of Health Psychology,, 29(09): 1367-1371.]].

2 Review of Existing Research on the Structure of "Emotional Labor Ability"

The concepts of surface acting, deep acting, and genuine emotion expression, as mentioned by experts, belong to "emotional labor strategies," which refer to the strategies or means employed by individuals in the process of emotional labor. These are not the structure of emotional labor itself.

Existing research has conducted preliminary explorations into the structural components of "emotional labor ability." In the early stages of research on emotional labor ability, some studies equated "emotional intelligence" with "emotional labor ability" based on the interpersonal interaction characteristics of emotional labor[[[]Mayer, J. D., & Salovey, P.. (1993). The intelligence of emotional intelligence. Imagination Cognition and Personality, 17(3), 433-442. DOI:10.1016/0160-2896(93)90010-3.]], proposing that "emotional labor ability primarily encompasses aspects such as emotional perception ability, emotional understanding ability, emotional management ability, and emotional utilization ability."[[[]Mayer, J. D., Salovey, P., Caruso, D. R., & Sitarenios, G.. (2003). Measuring emotional intelligence with the msceit v2.0. Emotion, 3(1), 97-105. DOI:10.1037/1528-3542.3.1.97.]][[[]Mayer, J. D., Salovey, P., & Caruso, D. R. (2004). Emotional intelligence: Theory, findings, and implications. Psychological Inquiry, 15(3), 197e215. https://doi.org/10.1207/s15327965pli1503_02.]][[[]Jennings, P. A., & Greenberg, M. T. (2009). The prosocial classroom: Teacher social and emotional competence in relation to student and classroom outcomes. Review of Educational Research, 79(1), 491e525. https://doi.org/10.3102/0034654308325693.]]

Specifically, focusing on the teacher population, researchers have indicated that teachers' emotional labor ability consists of three components: emotional perception ability, emotional regulation ability, and emotional expression ability[[[]Rindermann, H. (2009). Ekf: Emotionale-Kompetenz-Fragebogen. Hogrefe.]]. Jennings et al. (2013)[[[]Jennings, P. A., Frank, J. L., Snowberg, K. E., Coccia, M. A., & Greenberg, M. T. (2013). Improving classroom learning environments by Cultivating Awareness and Resilience in Education (CARE): Results of a randomized controlled trial. School Psychology Quarterly, 28(4), 374e390. https://doi.org/10.1037/spq0000035.]] proposed that the emotional skills teachers need to acquire include affective skills, mindfulness awareness, and caring and compassion. Based on the aforementioned research findings, it can be inferred that emotional intelligence is a crucial psychological resource for individuals to navigate emotional labor [[[]Wong, C. S., & Law, K. S.. (2002). The effects of leader and follower emotional intelligence on performance and attitude: an exploratory study. Leadership Quarterly, 13(3), 243-274.DOI:10.1016/S1048-9843(02)00099-1.]]. However, high emotional intelligence does not necessarily imply efficient emotional labor, as possessing a skill does not equate to successfully applying it.

As research on emotional labor ability progresses, researchers have begun to focus on exploring individuals' emotional labor ability within specific work contexts. It is believed that emotional labor ability should manifest as an individual's capacity to employ various emotional labor strategies, such as deep acting and genuine emotion expression, based on specific educational situations[[[]Wang, H., Hall, N. C., & Taxer, J. L. (2019). Antecedents and consequences of teachers' emotional labor: A systematic review and meta-analytic investigation. Educational Psychology Review, 1e36. https://doi.org/10.1007/s10648-019-09475-3.]][[[]Chang, M.-L., & Taxer, J. (2020). Teacher emotion regulation strategies in response to classroom misbehavior. Teachers and Teaching, 1e17. https://doi.org/10.1080/13540602.2020.1740198.]][[[]Yin, H., Huang, S., & Wang, W. (2016). Work environment characteristics and teacher well-being: The mediation of emotion regulation strategies. International Journal of Environmental Research and Public Health, 13(9), 907. https://doi.org/10.3390/ijerph13090907.]]. In the context of kindergarten teachers, existing research has summarized experiences and presented viewpoints on the conceptual connotation[[[]Hong, S.. (2018). The Study on the Teacher's Emotional Labor in Early Childhood Education Institution. DONGAINMUNHAK, 45,473-494.DOI:10.52639/JEAH.2018.12.45.473.]], importance[[[]Zhang, KS., Cui, XX., Wang, RD., Mu, CC., Wang, F.. (2022). Emotions, Illness Symptoms, and Job Satisfaction among Kindergarten Teachers: The Mediating Role of Emotional Exhaustion. SUSTAINABILITY, 14(6), 3261. DOI:10.3390/su14063261.]], and improvement strategies of kindergarten teachers' emotional labor ability[[[]Seom, H S.. (2022). An Analysis on Kindergarten Teachers’ Emotional Labor. The Journal of EducationKyungin University of Education Educational Research Institute,42(3),137-151. DOI:10.25020/je.2022.42.3.137.]]. This provides valuable insights for further exploring the structural dimensions of kindergarten teachers' emotional labor ability.

3 Theoretical Foundation and the Five Dimensions of "Emotional Labor Ability" Proposed in This Study

The model of emotional labor mechanisms proposed by Grandey is closely linked to the five dimensions of "emotional labor ability" put forth in this study and provides a solid theoretical foundation for the development of the "Emotional Labor Ability Scale for Kindergarten Teachers."

Grandey's model divides the working mechanism of emotional labor into three stages: situational cues, emotional regulation processes, and long-term outcomes[[[]Grandey, A. (2000). Emotional regulation in the workplace: a new way to conceptualize emotional labor. Journal of Occupational Health Psychology, 5(1), 95-110. DOI:10.1037/1076-8998.5.1.95.]]. Specifically, Grandey emphasizes that during the stage of situational cues, individuals must first use emotional intelligence to perceive and understand the interaction expectations of work objects and the attributes of emotional events, and have a correct understanding and deep identification with the rules of emotional labor, to ensure that subsequent emotional expressio

---

## [Decision Letter · Decision Letter 1]

Dear Dr. Hong,

Thank you for submitting your manuscript to PLOS ONE. After careful consideration, we feel that it has merit but does not fully meet PLOS ONE’s publication criteria as it currently stands. Therefore, we invite you to submit a revised version of the manuscript that addresses the points raised during the review process.

We kindly ask you to revisit the PLOS ONE author guidelines to confirm adherence to all necessary requirements.

Please submit your revised manuscript within May 15 2025 11:59PM. If you will need more time than this to complete your revisions, please reply to this message or contact the journal office at plosone@plos.org . A rebuttal letter that responds to each point raised by the academic editor and reviewer(s). You should upload this letter as a separate file labeled 'Response to Reviewers'.A marked-up copy of your manuscript that highlights changes made to the original version. You should upload this as a separate file labeled 'Revised Manuscript with Track Changes'.An unmarked version of your revised paper without tracked changes. You should upload this as a separate file labeled 'Manuscript'.

We look forward to receiving your revised manuscript.

Kind regards,

Mohamed Ahmed Said, Ph.D.

Academic Editor

PLOS ONE

Additional Editor Comments:

Abstract

Please expand "BEIs" to its full form in the abstract for clarity, especially for readers unfamiliar with the acronym.

Introduction

Revise lines 7-10 in the second paragraph for clarity and conciseness.

The introduction discusses multiple related constructs (emotional labor, emotional intelligence, emotional labor ability) but does not clearly differentiate them. Please provide a more precise definition of "emotional labor ability" early on to strengthen conceptual clarity.

Clarify why existing tools are inadequate or unsuitable for kindergarten teachers specifically in the introduction.

Some sections, particularly the discussion of emotional intelligence, could be more concise. Maintain focus on emotional labor ability as a distinct construct.

Methods

Statistical Analysis Details: Provide a detailed description of the statistical methods used, including the software, tests, assumptions, treatment of outliers, missing data handling, and sample size calculations. This will improve transparency and reproducibility.

Participant Demographics: Include detailed demographic information about the participants (age, gender distribution, years of experience, etc.). This will clarify the sample characteristics and support generalizability.

Data Availability Statement: Include a data availability statement specifying where the data will be made available, following PLOS ONE’s data sharing policies.

Provide an ethics approval number or reference ID to comply with PLOS ONE’s ethical standards.

Ensure consistency in capitalization in Figure 1. Standardize terms and follow PLOS ONE’s formatting rules for figures.

The sample of 4 female teachers and 1 male teacher may not be sufficient to explore the unique qualities of emotional labor ability. Could you explain how this small sample was justified?

For the Delphi Method, please provide the characteristics (age, gender, specialization, experience, etc.) of the experts involved.

Figure 2: Use title case for all headings/subheadings (e.g., "Emotional Recognition" instead of "emotional recognition") to follow academic conventions.

Questionnaire Survey: The first 4 lines of the survey are duplicated. Please remove the redundancy.

Sampling and Representativeness

Ensure the sample accurately reflects the broader population of kindergarten teachers. The convenience sampling method was used, please clarify how this may impact the generalizability of the results.

Clarify whether any additional criteria (e.g., age, gender, years of experience) were considered in participant selection. How did you ensure diversity and representativeness in the sample for scale development?

Although 818 valid responses were collected, more information on how the sample size was determined and whether statistical power analysis was conducted for factor analyses would be helpful. Was the sample size adequate for these analyses, and how does it support the scale’s validity?

Demographics and Scale Details

Provide more information on participants’ demographics (e.g., gender, age, years of experience, educational background). This will help assess whether the sample is diverse enough for developing a valid Emotional Labor Ability Scale.

Please specify how many items each of the five dimensions in the questionnaire contains.

Clarify whether the questionnaire includes negatively worded items that require reverse scoring and explain how these items are handled in the final scoring process. This will ensure transparency in how emotional labor ability is quantified.

Results

The authors have not provided information regarding participant scores. This could be an oversight, as such details are crucial for understanding the results and ensuring transparency in the data analysis process. Please clarify how participant scores were handled and presented.

In the Excel spreadsheet, it is unclear what "Q59" refers to, as the manuscript states that the questionnaire consists of 37 items across five dimensions. Could the authors clarify whether this is an error, an omitted item, or if additional questions were used but not reported in the manuscript?

It would be beneficial to include a copy of the questionnaire with the manuscript, either as a supplementary file or by presenting all items in full within a table. This would enhance transparency and allow readers to assess the scale's content and structure.

Reviewers' comments:

Reviewer's Responses to Questions

**Comments to the Author**

Reviewer #1: (No Response)

Reviewer #3: All comments have been addressed

Reviewer #4: (No Response)

2. Is the manuscript technically sound, and do the data support the conclusions?

Reviewer #1: Yes

Reviewer #3: Yes

Reviewer #4: Partly

3. Has the statistical analysis been performed appropriately and rigorously?

Reviewer #1: Yes

Reviewer #3: Yes

Reviewer #4: Yes

4. Have the authors made all data underlying the findings in their manuscript fully available?

Reviewer #1: Yes

Reviewer #3: Yes

Reviewer #4: Yes

5. Is the manuscript presented in an intelligible fashion and written in standard English?

Reviewer #1: Yes

Reviewer #3: Yes

Reviewer #4: Yes

Reviewer #1: I am totally satisfied with the article and the changes. Only one more little change is needed:

[2]Beauchamp, & Catherine. (2015). Reflection in teacher education: issues emerging from a review o f current literature. Reflective Practice, 16(1), 123-141. https://doi.org/10.1080/14623943.2014.98252 5.

should be written as

[2]Beauchamp, C. (2015). Reflection in teacher education: issues emerging from a review of current literature. Reflective Practice, 16(1), 123-141. https://doi.org/10.1080/14623943.2014.98252 5

Reviewer #3: Thank you for addressing all the required issues from the first review. In order to make the article clearer, the authors should provide a clearer comparison with existing emotional labour-related scales and their limitations to address the lack of appropriate tools for measuring EL in kindergarten teachers.

In the practical implication, a brief discussion on how the scale could be integrated into actual kindergarten teacher training programmes and how its effectiveness could be tested in future studies.

Reviewer #4: In general, the authors have coped with the task of developing a scale of Emotional Labor Ability of Kindergarten Teachers and proposed their own version of operationalisation for the Emotional Labor construct and developed their scale in accordance with it. Of course, it would be possible to discuss the operationalisation option. For example, why only such components of emotional intelligence as recognition and understanding are considered as individual factors, but such component as empathy is not included. However, in this case we can agree with the authors' understanding of this construct.

Psychometric verification of the developed scale was carried out quite correctly, but the results of confirmatory analysis, resulting in the proposed five-factor model, do not seem to be complete, as the authors present (Table 3) fit indices only for one- and five-factor models. T he question about the quality of models with other number of factors (2, 3 ,4?) remains unanswered. If the authors had presented data on the basis of which the readers would have been convinced of the advantage of the five-factor model, then the work of the researchers would have been completed and substantiated.

As for the correctness of the use of language, it is mostly correct, however, I am not sure about the capital W in the sentence ‘...that is, When there is a certain gap between the kindergarten teachers’ inner emotional feelings ....’ (p.18).

**Do you want your identity to be public for this peer review?** For information about this choice, including consent withdrawal, please see our Privacy Policy

Reviewer #1: No

Reviewer #3: No

Reviewer #4: No

---

## [Author Response · Author response to Decision Letter 2]

16 Apr 2025

Report

Reviewer1 13.04.2025

Abstract

1.Please expand "BEIs" to its full form in the abstract for clarity, especially for readers unfamiliar with the acronym.

Thanks for the comment. Based on the comment, I have made the following modifications to the abstract:

Second, Behavioral Event Interview were conducted with 21 teachers to construct primary and secondary competency elements.

Introduction

2.Revise lines 7-10 in the second paragraph for clarity and conciseness.

Thanks for the comment. Based on the comment, I have condensed the content of the paper:

In terms of the teacher group, some researchers have pointed out that teachers' emotional labor ability includes three components: emotional perception ability, emotional regulation ability, and emotional expression ability.

3.The introduction discusses multiple related constructs (emotional labor, emotional intelligence, emotional labor ability) but does not clearly differentiate them. Please provide a more precise definition of "emotional labor ability" early on to strengthen conceptual clarity.

Thanks for the comment. Based on the comment, I have added the definition of "emotional labor ability of kindergarten teachers" in the corresponding position of the introduction:

Based on existing research results, this study believes that the emotional labor ability of kindergarten teachers refers to the ability of kindergarten teachers to actively regulate their internal emotional experiences and external emotional expressions according to the professional requirements of kindergarten teachers in the process of education and teaching, in order to complete educational and teaching tasks.

4.Clarify why existing tools are inadequate or unsuitable for kindergarten teachers specifically in the introduction.

Thanks for the comment. Based on the comment, I have provided explanations in the corresponding position of the introduction:

Based on the suggestion, I have added the following content to the introduction section of the article:

The existing research results have provided useful references for us to understand the concept of emotional labor ability of kindergarten teachers. However, we need to recognize that the existing research mainly adopts speculative research methods to empirically explore the constituent elements of emotional labor ability of kindergarten teachers, and has not yet adopted empirical research methods to systematically verify and demonstrate the structural dimensions of emotional labor ability of kindergarten teachers. Based on this, there is still significant room for exploration in terms of theoretical appropriateness and practical explanatory power of existing research conclusions on the emotional labor ability of kindergarten teachers. This research gap provides an important opportunity for theoretical innovation and methodological breakthroughs in this study.

5.Some sections, particularly the discussion of emotional intelligence, could be more concise. Maintain focus on emotional labor ability as a distinct construct.

Thanks for the comment. Based on the comment, I have removed this paragraph from the introduction section of the article:

Based on the above research results, it is speculated that emotional intelligence is an important psychological resource for individuals to cope with emotional labor, but high emotional intelligence does not necessarily mean efficient emotional labor, because having ability is not equivalent to being able to successfully apply it.

Methods

6.Statistical Analysis Details: Provide a detailed description of the statistical methods used, including the software, tests, assumptions, treatment of outliers, missing data handling, and sample size calculations. This will improve transparency and reproducibility.

Thanks for the comment. Based on the comment, I have added an explanation of the analysis software in the corresponding section of the methods

After collecting valid data, use SPSS software to randomly split the data into two parts.

7.Participant Demographics: Include detailed demographic information about the participants (age, gender distribution, years of experience, etc.). This will clarify the sample characteristics and support generalizability.

Thank you for your careful review and valuable suggestions on this article. We fully agree with the importance of your request to supplement demographic variable characteristics. Demographic information does help to more comprehensively evaluate the applicability of the scale among different teacher groups and provide more targeted references for subsequent research.

In this study, our primary goal was to develop and validate the psychometric characteristics of the scale (such as construct validity and reliability), therefore demographic data was not systematically collected. This design choice is mainly based on two considerations: (1) focusing on the validity testing of the core construct of the scale; (2) Control the length of the questionnaire to improve the completion rate of teachers' responses. We acknowledge that the lack of demographic variables does limit the analysis of group differences, but it should be noted that the current reliability and validity test results of the scale are based on representative overall data of the teacher population, and their scientific validity is not affected.

To actively respond to your suggestions, we plan to take the following improvement measures in research:

In the Limitations and Future Research section of this article, we provide an explanation of the current limitations and suggest that future research further explore the impact of demographic characteristics on emotional labor capacity.

Fourth, In this study, our primary goal was to develop and validate the psychometric characteristics of the scale (such as construct validity and reliability), therefore demographic data was not systematically collected. In the next stage of the large sample survey, we will include core demographic indicators (such as age, teaching experience, education, kindergarten type, etc.) and analyze the relationship between these factors and scale scores.

support generalizability

8.Data Availability Statement: Include a data availability statement specifying where the data will be made available, following PLOS ONE’s data sharing policies.

Thanks for the comment. This study mainly used data obtained from a questionnaire survey, which has been included in the "Supporting Information" file in the form of an Excel spreadsheet. When submitting the revised paper, it was also submitted to the submission system for readers to use.

In addition, I provided the following explanation in my paper:

Data availability

The data used in this study has been included in the 'Supporting Information' file and can be published together with the paper for readers to use.

9.Provide an ethics approval number or reference ID to comply with PLOS ONE’s ethical standards.

Thanks for the comment. I have indicated the ethical approval number in the article and submitted the original ethical approval document as an attachment to the system for your reference.

The ethical approval number is 2023255.

10.Ensure consistency in capitalization in Figure 1. Standardize terms and follow PLOS ONE’s formatting rules for figures.

Thanks for the comment. Based on the comment, I have adjusted the capitalization of the text in the image based on the suggestions and uploaded the relevant images as attachments.

11.The sample of 4 female teachers and 1 male teacher may not be sufficient to explore the unique qualities of emotional labor ability. Could you explain how this small sample was justified?

Thanks for the comment.

Based on the advice of the previous external audit expert, we have supplemented the interviews with 18 female teachers and 3 male teachers. We have updated the information in the abstract, but we forgot to update the interviewee's information here. Please forgive our negligence. I have now made a correction in the article:

Among them, there were 18 female teachers and 3 male teacher.

12.For the Delphi Method, please provide the characteristics (age, gender, specialization, experience, etc.) of the experts involved.

Thanks for the comment. Based on the comment, I inserted a table in the corresponding position of the article listing all the information of the experts.

Their demographic variable characteristics are shown in Table 1.

Table 1 Demographic Characteristics of the Experts

Characteristics Frequency (N) Proportion (%)

Age (years)

30-40 1 20%

40-50 3 60%

≥50 1 20%

Highest educational degree

College 0 0%

Undergraduate 1 20%

Master’s and doctorate 4 80%

Title

Junior professional title 1 20%

Senior professional title 2 40%

Associate senior professional title 2 40%

Work experience (years)

11–15 3 60%

15–20 1 20%

≥20 1 20%

Research field

Kindergarten teacher education 2 40%

Kindergarten Teacher Management 2 40%

Basic Theory of Preschool Education 1 20%

13.Figure 2: Use title case for all headings/subheadings (e.g., "Emotional Recognition" instead of "emotional recognition") to follow academic conventions.

Thanks for the comment. Based on the comment, I have adjusted the letters in the image and uploaded it as an attachment.

14.Questionnaire Survey: The first 4 lines of the survey are duplicated. Please remove the redundancy.

Thanks for the comment. Based on the comment, I have already deleted the corresponding position in the text as requested.

Sampling and Representativeness

15.Ensure the sample accurately reflects the broader population of kindergarten teachers. The convenience sampling method was used, please clarify how this may impact the generalizability of the results.

Thanks for the comment. Based on the comment, I have provided the following explanation at the corresponding position in the text:

Second, Due to my limited personal abilities, I used convenience sampling to conduct a questionnaire survey and reliability and validity testing only among kindergarten teachers in Hunan Province, China. The sample size may not represent the different urban and rural areas and regions of kindergarten teachers in China, and future research needs to further expand the survey population.

16.Clarify whether any additional criteria (e.g., age, gender, years of experience) were considered in participant selection. How did you ensure diversity and representativeness in the sample for scale development?

Thanks for the comment.

During the research process, I made efforts to collect more questionnaires, but did not screen demographic variables such as age, gender, and teaching experience of the participants. Instead, I conducted reliability and validity tests on the data after the survey to ensure that the scale had good psychological measurement characteristics.

17.Although 818 valid responses were collected, more information on how the sample size was determined and whether statistical power analysis was conducted for factor analyses would be helpful. Was the sample size adequate for these analyses, and how does it support the scale’s validity?

Thanks for the comment. Based on the comment, I have provided the following explanation on the issue of sample size in the relevant section of the paper:

Previous studies have suggested that the number of valid questionnaires should be at least 10 times the number of items on the scale to be more reliable (He, 2025). The number of valid questionnaires in this study is 22.1 times the total number of questions, which is much larger than the theoretical sample threshold (37 × 10). Therefore, it meets the sample conditions of the questionnaire survey and reflects the richness of the data volume.

Demographics and Scale Details

18. Provide more information on participants’ demographics (e.g., gender, age, years of experience, educational background). This will help assess whether the sample is diverse enough for developing a valid Emotional Labor Ability Scale.

Thank you for your careful review and valuable suggestions on this article. We fully agree with the importance of your request to supplement demographic variable characteristics. Demographic information does help to more comprehensively evaluate the applicability of the scale among different teacher groups and provide more targeted references for subsequent research.

In this study, our primary goal was to develop and validate the psychometric characteristics of the scale (such as construct validity and reliability), therefore demographic data was not systematically collected. This design choice is mainly based on two considerations: (1) focusing on the validity testing of the core construct of the scale; (2) Control the length of the questionnaire to improve the completion rate of teachers' responses. We acknowledge that the lack of demographic variables does limit the analysis of group differences, but it should be noted that the current reliability and validity test results of the scale are based on representative overall data of the teacher population, and their scientific validity is not affected.

To actively respond to your suggestions, we plan to take the following improvement measures in future research:

In the next stage of the large sample survey, we will include core demographic indicators (such as age, teaching experience, education, kindergarten type, etc.) and analyze the relationship between these factors and scale scores.

In the discussion section of this article, we will provide an explanation of the current limitations and suggest that future research further explore the impact of demographic characteristics on emotional labor capacity.

developing a valid Emotional Labor Ability Scale.

19.Please specify how many items each of the five dimensions in the questionnaire contains.

Thanks for the comment. Based on the comment, I have provided the following explanations in the relevant sections of the article:

Emotional intelligence refers to the ability of kindergarten teachers to recognize and understand the emotional state of others in interpersonal interaction, and to use this information to guide the whole process of emotional labor. It includes two sub-dimensions, namely the ability of emotional recognition and the ability of emotional understanding, and a total of 8 items.

The ability of internalizing emotional labor rules refers to the ability of kindergarten teachers to deeply identify with and consciously practice the code of conduct of emotional labor for the normal operation of emotional labor, including the subconsciousness of emotional labor rules and the automation of emotional labor rules, with a total of 5 items.

The coordination ability in emotional labor refers to the ability of kindergarten teachers to deal with emotional events in complex educational situations while ensure the smooth progress of all aspects of the kindergarten's daily activities. It mainly includes six sub-dimensions, namely the ability to coordinate collective teaching activities of the kindergarten, the ability to coordinate the morning activities of children, the ability to coordinate the life activities of children, the ability to coordinate the activities in corresponding activity area, and the ability to coordinate the outdoor or after-school activities of children, totaling 6 items.

The application ability to emotional labor strategies refers to the ability of kindergarten teachers to use certain emotional labor strategies according to the needs of educational situations in the process of emotional labor, including two sub-dimensions, namely the ability to use deep performance strategy and the ability to use natural performance strategy, with a total of 6 items.

The essence of reflective ability after emotional labor is that kindergarten teachers actively review their own emotional labor process to find their own problems in emotional labor and to find effective ways to solve problems, including three sub-dimensions, namely reflective perseverance, the ability to reflect on relevant theoretical knowledge and the ability to reflect on emotional labor experience, with a total of 8 items.

20.Clarify whether the questionnaire includes negatively w

---

## [Editor Report · Decision Letter 2]

Dear Dr. Hong,

Please submit your revised manuscript within Jun 13 2025 11:59PM. If you will need more time than this to complete your revisions, please reply to this message or contact the journal office at plosone@plos.org . A rebuttal letter that responds to each point raised by the academic editor and reviewer(s). You should upload this letter as a separate file labeled 'Response to Reviewers'.A marked-up copy of your manuscript that highlights changes made to the original version. You should upload this as a separate file labeled 'Revised Manuscript with Track Changes'.An unmarked version of your revised paper without tracked changes. You should upload this as a separate file labeled 'Manuscript'.

We look forward to receiving your revised manuscript.

Kind regards,

Mohamed Ahmed Said, Ph.D.

Academic Editor

PLOS ONE

Additional Editor Comments:

Editorial Evaluation Report on Outstanding Issues

Comment 16: Clarify whether any additional criteria (e.g., age, gender, years of experience) were considered in participant selection. How did you ensure diversity and representativeness in the sample for scale development?

While the authors' response is methodologically acceptable in focusing on psychometrics, the absence of demographic information limits the generalizability and interpretability of the scale. The explanation does not fully address the issue of sample representativeness. Since the construct relates to emotional labor in teachers, factors such as gender, teaching experience, and age are highly relevant contextual moderators.

Recommendation:

Authors should include a justification for the representativeness of their sample based on available institutional or population data and explicitly address this limitation in the manuscript. Additionally, future data collection plans to incorporate demographic diversity should be clearly integrated into the manuscript, not merely mentioned in the response letter.

Comment 18: Provide more information on participants’ demographics (e.g., gender, age, years of experience, educational background) to assess whether the sample is diverse enough for developing a valid Emotional Labor Ability Scale.

The authors' response does not address the core issue: without a demographic breakdown, the applicability of the scale across subgroups remains unknown. Furthermore, the current manuscript provides no empirical evidence that the sample is representative.

Recommendation:

The authors must clearly state in the manuscript that the generalizability of their results is limited due to the lack of demographic information and propose a concrete plan for stratified validation in future research stages.

Comment 20: Clarify whether the questionnaire includes negatively worded items that require reverse scoring and explain how these items are handled in the final scoring process.

The authors explicitly state that no reverse-scored items were included; however, this is inconsistent with the wording of several items, which are negatively phrased and logically require reverse scoring to align with the intended scale directionality (i.e., higher scores reflecting stronger emotional labor ability).

Upon review, the following items appear negatively worded and may require reverse coding:

• Item 9: "For me, the normative requirements that kindergarten teachers need to follow when expressing their emotions are not unfamiliar."

→ The double negative structure introduces ambiguity; it may function as a reverse item depending on interpretation.

• Item 17: "When interacting with young children, I can naturally show a sad emotional state to meet work needs (such as reaching empathy with young children)."

→ Agreement may indicate positive ability, but the interpretation could vary.

• Item 18: "When interacting with young children, I can naturally show an angry emotional state to meet work needs (such as expressing negation of problematic behavior in young children)."

→ Similar concerns apply regarding the positive/negative framing.

• Item 19: "When interacting with young children, I can naturally show a tense emotional state to meet work needs (such as attracting children's attention, etc.)."

→ The framing may confuse respondents without clarification.

• Item 32: "When my mood is relatively low, I can still persist in reflecting on whether my emotional expression behavior is reasonable."

→ Despite positive intent, the "low mood" condition could affect responses.

Recommendation:

To ensure scoring accuracy and transparency:

1. Review and identify all items with negative or ambiguous phrasing, especially those listed above.

2. Clearly indicate in the manuscript whether these items were reverse-coded or not. If not, discuss the rationale and implications for validity.

3. Consider rewording these items in future iterations to avoid semantic confusion (especially Item 9).

4. Include a scoring guide (e.g., a coding table) as an appendix or supplementary file to facilitate replicability and future validation studies.

Failure to handle reverse-coded items properly could distort factor structures, produce misleading total scores, and lead to misinterpretations of emotional labor profiles.

Comment 21: Clarify how participant scores were handled and presented.

Although the methodology appears reasonable, the presentation of results is incomplete. The manuscript does not report descriptive statistics (means, standard deviations, ranges) for total or subscale scores. This omission limits the reader's ability to interpret the scale’s effectiveness and score distribution.

Recommendation:

The authors should report descriptive statistics (mean, SD, minimum, maximum) for total and subscale scores within the manuscript to allow readers to evaluate the scale’s distributional properties and interpretability.

Inconsistency in Number of Items Reported

Observation:

In the manuscript, the authors refer to a 39-item scale, whereas the submitted version for review contains only 33 items.

Recommendation:

The authors must:

1. Clarify the final number of items on the validated scale.

2. Explain whether any items were excluded during factor analysis.

3. Include a complete, numbered list of the final items in an appendix or supplementary material to ensure transparency.

---

## [Author Response · Author response to Decision Letter 3]

30 Apr 2025

Report

Additional Editor Comments 30.04.2025

Comment 16: Clarify whether any additional criteria (e.g., age, gender, years of experience) were considered in participant selection. How did you ensure diversity and representativeness in the sample for scale development?

While the authors' response is methodologically acceptable in focusing on psychometrics, the absence of demographic information limits the generalizability and interpretability of the scale. The explanation does not fully address the issue of sample representativeness. Since the construct relates to emotional labor in teachers, factors such as gender, teaching experience, and age are highly relevant contextual moderators.

Recommendation:

Authors should include a justification for the representativeness of their sample based on available institutional or population data and explicitly address this limitation in the manuscript. Additionally, future data collection plans to incorporate demographic diversity should be clearly integrated into the manuscript, not merely mentioned in the response letter.

Comment 18: Provide more information on participants’ demographics (e.g., gender, age, years of experience, educational background) to assess whether the sample is diverse enough for developing a valid Emotional Labor Ability Scale.

The authors' response does not address the core issue: without a demographic breakdown, the applicability of the scale across subgroups remains unknown. Furthermore, the current manuscript provides no empirical evidence that the sample is representative.

Recommendation:

The authors must clearly state in the manuscript that the generalizability of their results is limited due to the lack of demographic information and propose a concrete plan for stratified validation in future research stages.

Thanks for the comment. Due to the correlation between the two comments mentioned above, I will integrate them and provide a unified response.

The demographic variable characteristics of the target group are one of the factors for verifying sample representativeness. At the same time, we need to note that there are other factors that can ensure the representativeness of the sample. For example, previous studies have shown that survey data with large sample sizes, diverse sample regions, and random sampling methods can improve sample representativeness and reliability, making research results more likely to approach the true characteristics of the population[Xu S.C., Sun Q.. (2025). The Influence of Family Social Capital on College Students' Fertility Intention. Journal of Social Sciences, Jilin University, 65 (02): 181-195+239. https://doi.org/10.15939/j.jujsse.2025.02.sh2.][Hao Y.W., Jin Y.J.. (2020). Design of Spatial Balanced Sampling with Geographic Coordinate Information Participation. Mathematical Statistics and Management, 39 (06): 978-989. https://doi.org/10.13860/j.cnki. sltj. 2020, 1025-001.].

In the specific context of this study,The number of valid questionnaires in this study is 22.1 times the total number of questions, which is much larger than the theoretical sample threshold (37 × 10). Moreover, this study to some extent considered the geographical distribution of data sources. For the convenience of personal work and study, this study mainly randomly distributed questionnaires to Hunan Province, Xinjiang Uyghur Autonomous Region, and Jiangsu Province. Among them, Hunan Province distributed 366 questionnaires, Xinjiang Uyghur Autonomous Region distributed 158 questionnaires, and Jiangsu Province distributed 294 questionnaires. The sample size has covered the eastern, central, and western regions of China. In summary, this study demonstrated a high degree of representativeness and diversity in the research data due to its large sample size, extensive geographical coverage, and random sampling method.

At the same time, despite my great efforts in distributing the questionnaire, I was unable to balance the proportion of demographic variables among the survey subjects due to the limited personal abilities of the researchers. Therefore, although the scale has been formed based on solid theoretical foundations, two rounds of expert opinions, interviews with grassroots kindergarten teachers, and a large amount of questionnaire survey data, it still needs to expand the sample size and empirical research scope in the future to enhance its representativeness and applicability.

Based on the comment, I have made the following modifications to the paper:

2.2.4 Questionnaire Survey

The questionnaire comprises five dimensional elements and 37 items, with each question item adopting the Likert five-point scoring method to facilitate quantitative analysis. Previous studies have shown that survey data with large sample sizes, diverse sample regions, and random sampling methods can improve sample representativeness and reliability, making research results more likely to approach the true characteristics of the population (Xu & Sun, 2025; Hao & Jin , 2020). Therefore, in order to improve the representativeness and diversity of the sample, This study mainly aimed to randomly distribute questionnaires to in-service kindergarten teachers in Hunan Province, Xinjiang Uygur Autonomous Region, and Jiangsu Province, with a total of 818 valid questionnaires collected. Among them, Jiangsu Province collected 294 valid questionnaires, Hunan Province collected 366 valid questionnaires, and Xinjiang Uygur Autonomous Region collected 158 valid questionnaires. The sample size has covered the eastern, central, and western regions of China. In addition, Previous studies have suggested that the number of valid questionnaires should be at least 10 times the number of items on the scale to be more reliable (He, 2025). The number of valid questionnaires in this study is 22.1 times the total number of questions, which is much larger than the theoretical sample threshold (37 × 10). In summary, this study demonstrates high representativeness and diversity with its large sample size, extensive geographical coverage, and random sampling method. After collecting valid data, use SPSS software to randomly split the data into two parts. The first 409 pieces of data were used for exploratory factor analysis (EFA), which aimed to identify the underlying factors or dimensions of the questionnaire. The other 409 pieces of data were reserved for confirmatory factor analysis (CFA), which was conducted to verify the factor structure identified in the EFA and assess the questionnaire's construct validity. Overall, this study employed several analytical methods including item analysis, exploratory factor analysis, confirmatory factor analysis, reliability and Validity analysis.

5. Limitations and Future Research

Second, although the scale has been formed based on solid theoretical foundations, two rounds of expert opinions, interviews with grassroots kindergarten teachers, and a large amount of questionnaire survey data, research results also indicate that the scale has good psychological measurement characteristics. However, the lack of demographic variable information may lead to limited universality of research results. Therefore, in the future stage of large sample surveys, we will include core demographic indicators such as age, teaching experience, education, kindergarten type, etc. for further in-depth exploration to improve the representativeness and universality of research results.

I have added the new literature introduced in this revision to the reference list:

[15]Hao Y.W., Jin Y.J.. (2020). Design of Spatial Balanced Sampling with Geographic Coordinate Information Participation. Mathematical Statistics and Management, 39 (06): 978-989. https://doi.org/10.13860/j.cnki. sltj. 2020, 1025-001.

[40]Xu S.C., Sun Q.. (2025). The Influence of Family Social Capital on College Students' Fertility Intention. Journal of Social Sciences, Jilin University, 65 (02): 181-195+239. https://doi.org/10.15939/j.jujsse.2025.02.sh2.

Comment 20: Clarify whether the questionnaire includes negatively worded items that require reverse scoring and explain how these items are handled in the final scoring process.

The authors explicitly state that no reverse-scored items were included; however, this is inconsistent with the wording of several items, which are negatively phrased and logically require reverse scoring to align with the intended scale directionality (i.e., higher scores reflecting stronger emotional labor ability).

Upon review, the following items appear negatively worded and may require reverse coding:

• Item 9: "For me, the normative requirements that kindergarten teachers need to follow when expressing their emotions are not unfamiliar."

→ The double negative structure introduces ambiguity; it may function as a reverse item depending on interpretation.

• Item 17: "When interacting with young children, I can naturally show a sad emotional state to meet work needs (such as reaching empathy with young children)."

→ Agreement may indicate positive ability, but the interpretation could vary.

• Item 18: "When interacting with young children, I can naturally show an angry emotional state to meet work needs (such as expressing negation of problematic behavior in young children)."

→ Similar concerns apply regarding the positive/negative framing.

• Item 19: "When interacting with young children, I can naturally show a tense emotional state to meet work needs (such as attracting children's attention, etc.)."

→ The framing may confuse respondents without clarification.

• Item 32: "When my mood is relatively low, I can still persist in reflecting on whether my emotional expression behavior is reasonable."

→ Despite positive intent, the "low mood" condition could affect responses.

Recommendation:

To ensure scoring accuracy and transparency:

1. Review and identify all items with negative or ambiguous phrasing, especially those listed above.

2. Clearly indicate in the manuscript whether these items were reverse-coded or not. If not, discuss the rationale and implications for validity.

3. Consider rewording these items in future iterations to avoid semantic confusion (especially Item 9).

4. Include a scoring guide (e.g., a coding table) as an appendix or supplementary file to facilitate replicability and future validation studies.

Failure to handle reverse-coded items properly could distort factor structures, produce misleading total scores, and lead to misinterpretations of emotional labor profiles.

Thanks for the comment. The above items do not require reverse coding because the higher the score of the above items, the higher their emotional labor ability.

Item 9: "For me, the normative requirements that kindergarten teachers need to follow when expressing their emotions are not unfamiliar."

Kindergarten teachers are not unfamiliar with the normative requirements that need to be followed when expressing emotions. This sentence means that kindergarten teachers are familiar with the laws of emotional labor.

The higher the score of this question, the more familiar the kindergarten teacher is with the rules of emotional labor and the stronger their emotional labor ability.

Item 17: "When interacting with young children, I can naturally show a sad emotional state to meet work needs (such as reaching empathy with young children)."

Item 18: "When interacting with young children, I can naturally show an angry emotional state to meet work needs (such as expressing negation of problematic behavior in young children)."

Item 19: "When interacting with young children, I can naturally show a tense emotional state to meet work needs (such as attracting children's attention, etc.)."

Firstly, I need to point out that in the context of kindergarten education, emotional states such as sadness, anger, and tension are not entirely negative. On the contrary, in different types of educational activities such as picture book storytelling and outdoor sports activities, kindergarten teachers often need to express the above different types of emotions according to specific teaching situations to achieve corresponding teaching goals. Therefore, for the group of kindergarten teachers, naturally showing emotional states such as sadness, anger, and tension in front of young children is a very normal educational behavior, and to some extent, it is also a teaching method.

Secondly, the above three items mainly point to the fourth dimension of emotional labor ability: the ability to apply emotional labor strategies. The ability to apply emotional labor strategies refers to the ability of kindergarten teachers to use certain emotional labor strategies according to the needs of educational contexts during the emotional labor process, including two sub dimensions: the ability to apply deep performance strategies and the ability to apply natural performance strategies. Among them, the ability to apply natural expression strategies is a higher-level ability than the ability to apply deep expression strategies. The higher the scores of the above three items, the more likely kindergarten teachers are to naturally express appropriate emotional behaviors in complex and changing educational situations to meet organizational requirements, that is, the higher their ability to apply natural expression strategies.

Item 32: "When my mood is relatively low, I can still persist in reflecting on whether my emotional expression behavior is reasonable."

This question mainly points to the fifth dimension of emotional labor ability of kindergarten teachers: emotional labor reflection ability. In this study, the dimension of emotional labor reflection ability includes three sub dimensions: reflective perseverance, the ability to reflect based on relevant theoretical knowledge, and the ability to reflect based on emotional labor experience. Among them, when my emotions are relatively low, I can still persist in reflecting on whether my emotional expression behavior is reasonable, which precisely demonstrates the reflective perseverance of kindergarten teachers. The higher the score of this question, the stronger the reflective perseverance in the emotional labor reflection ability of kindergarten teachers.

To make the expression easier to understand, I have changed the wording of item 9 to:

I am familiar with the emotional expression norms that kindergarten teachers need to follow

Comment 21: Clarify how participant scores were handled and presented.

Although the methodology appears reasonable, the presentation of results is incomplete. The manuscript does not report descriptive statistics (means, standard deviations, ranges) for total or subscale scores. This omission limits the reader's ability to interpret the scale’s effectiveness and score distribution.

Recommendation:

The authors should report descriptive statistics (mean, SD, minimum, maximum) for total and subscale scores within the manuscript to allow readers to evaluate the scale’s distributional properties and interpretability.

Thanks for the comment.

In this study, our fundamental objective is to develop and validate the psychometric characteristics of the scale (such as structural validity and reliability), and we need to ensure that all research steps serve this fundamental objective. If we add the scores of kindergarten teachers on each dimension of the scale, on the one hand, too much information would blur the focus of this study. On the other hand, since this study does not provide demographic variable information, merely adding the scores of kindergarten teachers on each dimension of the scale would have little reference value. The above is my thinking on this issue for your reference. If you still think it is necessary to add it, I can do so later.

Inconsistency in Number of Items Reported

Observation:

In the manuscript, the authors refer to a 39-item scale, whereas the submitted version for review contains only 33 items.

Recommendation:

The authors must:

1. Clarify the final number of items on the validated scale.

2. Explain whether any items were excluded during factor analysis.

3. Include a complete, numbered list of the final items in an appendix or supplementary material to ensure transparency.

Thanks for the comment.

The numbe

---

## [Editor Report · Decision Letter 3]

Dear Dr. Hong,

We look forward to receiving your revised manuscript.

Kind regards,

Mohamed Ahmed Said, Ph.D.

Academic Editor

PLOS ONE

Additional Editor Comments:

Editorial Report: Evaluation of Authors' Responses to Reviewer Comments

1. Detailed Evaluation of Responses

1.1 Sampling Representativeness & Demographic Data (Comments 16, 18)

Reviewer Concerns:

• Discrepancy between claimed "random sampling" and described convenience sampling

• Absence of reported demographic data (gender, age, experience)

Authors' Response:

• Emphasized geographic diversity (3 regions) and large sample size (N=818)

• Acknowledged inability to balance demographics due to practical constraints

Editorial Assessment:

Key Issues:

1. Sampling Methodology:

• Contradiction between "random" claims and convenience sampling description

• Geographic coverage doesn't compensate for non-probability sampling limitations

2. Demographic Reporting:

• Critical omission of basic demographic characteristics

• No evidence of sample representativeness regarding gender, age, or experience

Recommendations:

• Explicitly clarify sampling as convenience-based

• Disclose available demographic data, even if imbalanced

• Discuss implications of sampling limitations

1.2 Questionnaire Survey Methodology (Section 2.2.4)

Strengths:

• Adequate sample size (N=818, 22.1:1 item ratio)

• Geographic diversity across three Chinese regions

• Methodologically sound EFA→CFA pipeline

• Appropriate use of Likert scales and statistical software

Concerns and Recommendations:

1. Sampling Claims:

• Resolve "random sampling" contradiction

• Clearly describe actual distribution method

2. Demographic Reporting:

• Add table with basic demographic breakdown

• Discuss potential sampling biases

3. Text Editing:

• Remove duplicated EFA/CFA description

• Improve conciseness

1.3 Reverse-Scored Items (Comment 20)

Reviewer Concerns:

• Ambiguous phrasing in Items 9, 17-19, 32

• Potential need for reverse scoring

Authors' Response:

• Maintained no reverse coding needed

• Reworded Item 9 for clarity

Editorial Assessment:

• Theoretical rationale for scoring direction is acceptable

• Remaining concerns about item interpretability

Recommendations:

• Add scoring appendix clarifying directionality

• Include limitation regarding potential respondent confusion

1.4 Descriptive Statistics (Comment 21)

Reviewer Concerns:

• Missing descriptive statistics (means, SDs, ranges)

Authors' Response:

• Argued against inclusion, citing focus concerns

Editorial Assessment:

• Omission is methodologically unacceptable

• Basic descriptive statistics are essential

Recommendations:

• Require addition of descriptive statistics table

• Consider rejection if not provided

1.5 Item Count Consistency

Reviewer Concerns:

• Discrepancy in reported items counts

Authors' Response:

• Clarified 37 initial items → 33 after EFA → 30 after CFA

Editorial Assessment:

• Explanation is adequate but requires:

o Full disclosure of final 30 items

o Justification for all item deletions

2. Overall Evaluation

Strengths:

• Appropriate sample size and geographic coverage

• Sound psychometric methodology

• Engagement with reviewer comments

Critical Deficiencies:

1. Unresolved sampling contradictions

2. Missing demographic data

3. Lack of descriptive statistics

4. Incomplete item documentation

Decision:

• Major Revisions Required (conditional acceptance):

1. Clarify sampling methodology

2. Add demographic information

3. Include descriptive statistics

4. Provide complete final scale items

• Timeline: 2 weeks for revisions

3. Final Notes

While the study shows methodological strengths, full transparency regarding sampling, demographics, and scale development is essential for publication.

Signed,

[Editor's Name]

[Journal Name]

Editorial Report: Assessment of Authors' Reactions to Reviewer Feedback

The editorial evaluation identifies four significant shortcomings that must be addressed prior to publication:

1. Unresolved sampling discrepancies between claimed random sampling and described convenience methods

2. Absence of critical demographic data (gender, age, experience)

3. Missing descriptive statistics for scale scores

4. Insufficient documentation of final scale items

To merit publication, the authors must complete these essential revisions within two weeks:

• Clarify sampling methodology with precise terminology

• Incorporate available demographic characteristics

• Report basic descriptive statistics (means, SDs, ranges …….)

• Provide complete documentation of the final 30-item scale

While the study demonstrates sound psychometric methodology, full transparency regarding these elements remains imperative for scientific rigor and reproducibility. Conditional approval is contingent on satisfactory resolution of these matters.

1. Comprehensive Assessment of Responses

1.1 Sampling Representativeness and Demographic Data (Comments 16, 18)

Reviewer Issues:

• Inconsistency between asserted "random sampling" and outlined convenience sampling

• Lack of provided demographic information (gender, age, experience)

Response from the Authors:

• Highlighted geographic variety across three locations and a substantial sample size of 818 participants.

• Recognized the difficulty to equilibrate demographics owing to practical limitations

Editorial Evaluation:

Principal Concerns:

1. Sampling Methodology:

• Discrepancy between assertions of "random" sampling and the description of convenience sampling

• Geographic coverage fails to mitigate the limitations inherent in non-probability sampling

2. Demographic Reporting:

• Significant absence of fundamental demographic attributes

• Lack of data demonstrating sample representativeness in terms of gender, age, or experience

Recommendations:

• Explicitly define the sample method as convenience-based

• Disclose the available demographic data, regardless of imbalance

• Examine the implications of sampling limits

1.2 Methodology for Questionnaire Survey (Section 2.2.4)

Advantages:

• Sufficient sample size (N=818, 22.1:1 item ratio)

• Geographic variety across three Chinese regions

• Methodologically robust EFA to CFA pipeline

• Proper utilization of Likert scales and statistical software

Issues and Suggestions:

1. Sampling Assertions:

• Address the inconsistency in "random sampling"

• Provide a detailed explanation of the real distribution methodology

2. Demographic Reporting:

• Incorporate a chart detailing fundamental demographic distribution

• Examine possible sampling biases

3. Text Editing:

• Eliminate redundant EFA/CFA descriptions

• Enhance brevity

1.3 Inversely Scored Items (Comment 20)

Reviewer Concerns:

• Unclear wording in Items 9, 17-19, 32

• Possible requirement for reverse scoring

Authors' Response:

• No reverse coding required

• Item 9 rephrased for clarity

Editorial Evaluation:

•The theoretical justification for the scoring direction is satisfactory.

• Outstanding issues regarding item interpretability

Suggestions:

• Incorporate a scoring addendum to elucidate directionality

• Address limitations about possible respondent confusion

1.4 Descriptive Statistics (Comment 21)

Reviewer Concerns:

• Lack of descriptive statistics (means, standard deviations, ranges)

Authors' Response:

• Opposed inclusion, noting issues with focus

Editorial Evaluation:

• Omission is methodologically indefensible

• Fundamental descriptive statistics are imperative

Recommendations:

• Mandate the inclusion of a table for descriptive statistics.

• Contemplate rejection if not supplied

1.5 Consistency of Item Count

Reviewer Concerns:

• Inconsistency in reported item counts

Authors' Response:

• Clarified 37 initial items, resulting in 33 after Exploratory Factor Analysis (EFA) and 30 after Confirmatory Factor Analysis (CFA).

Editorial Evaluation:

• Include the final 30-item scale in full (wording + subscales) as a table or appendix.

• Justify all item deletions

Advantages:

• Optimal sample size and regional representation

• Robust psychometric methods

• Interaction with reviewer feedback

---

## [Author Response · Author response to Decision Letter 4]

19 May 2025

Report

Additional Editor Comments 16.05.2025

1.Clarify sampling methodology

Recommendations:

Explicitly clarify sampling as convenience-based

Thank you for your interest in the sampling method of this study. The expressions of convenience sampling and random sampling mentioned in the article are not contradictory. Now, the application relationship between convenience sampling and random sampling is explained as follows:

① Convenient Sampling for Regional Selection

Based on the researcher's existing research cooperation networks in Xinjiang Uygur Autonomous Region (western region), Hunan Province (central region), and Jiangsu Province (eastern region), we used convenience sampling to determine the sample area. This design ensures that the sample covers the three major economic geographical zones of eastern, central, and western China, thereby enhancing the geographical representativeness of the sample.

② Random sampling of teacher groups

In the selected area, we distributed questionnaires to in-service kindergarten teachers through random sampling to ensure that each teacher has an equal probability of being included in the study and avoid human selection bias.

The statement in the paper is:

Therefore, based on the researcher's existing research cooperation networks in Xinjiang Uygur Autonomous Region (western region), Hunan Province (central region), and Jiangsu Province (eastern region), we used convenience sampling to determine the sample area. This design ensures that the sample covers the three major economic geographical zones of eastern, central, and western China, thereby enhancing the geographical representativeness of the sample. In the selected area, we distributed questionnaires to in-service kindergarten teachers through random sampling to ensure that each teacher has an equal probability of being included in the study and avoid human selection bias.

2.Add demographic information

Recommendations:

Disclose available demographic data, even if imbalanced

Thank you for your attention to the sample features. This study has collected gender information from participants, including 50 male kindergarten teachers and 768 female kindergarten teachers. Based on your suggestion, we have added the following content in the relevant section of the paper:

In the valid questionnaire, there were 50 male kindergarten teachers, accounting for 6.11% of the total number, and 768 female kindergarten teachers, accounting for 93.89% of the total number.

At the same time, it should be noted that this study, as the initial validity validation stage of the scale development, was mainly based on the core objectives of psychometric assessment and did not systematically collect other demographic variables such as age and teaching experience. We acknowledge that this limitation may affect the group inference validity of the results. Based on your previous suggestion, we have made the following clarification on the limitations of our research:

Second, although the scale has been formed based on solid theoretical foundations, two rounds of expert opinions, interviews with grassroots kindergarten teachers, and a large amount of questionnaire survey data, research results also indicate that the scale has good psychological measurement characteristics. However, the lack of demographic variable information may lead to limited universality of research results. Therefore, in the future stage of large sample surveys, we will include core demographic indicators such as age, teaching experience, education, kindergarten type, etc. for further in-depth exploration to improve the representativeness and universality of research results.

3.Include descriptive statistics

Recommendations:

Mandate the inclusion of a table for descriptive statistics.

Thank you for your attention to the descriptive statistics. Based on your suggestion, I have added descriptive analysis to the paper, as follows:

According to the analysis of Descriptive Statistics (Table 8), The average values of emotional intelligence, the ability of internalizing emotional labor rules, the coordination ability in emotional labor, the reflective ability after emotional labor, and the application ability to emotional labor strategies in each dimension are 3.94, 4.05, 4.00, 3.99, and 3.97, respectively. The mean values of all dimensions are significantly higher than the theoretical median, indicating that the surveyed group of kindergarten teachers generally self evaluate to have high emotional labor ability. The standard deviation of each dimension ranges from 0.54 to 0.61, and the variance ranges from 0.30 to 0.38, indicating a relatively low degree of data dispersion.

Table 8 Descriptive Statistics(N=818)

Factor range minimum value Maximum value average value standard deviation variance

A emotional intelligence 4.00 1.00 5.00 3.94 0.54 0.30

B the ability of internalizing emotional labor rules 4.00 1.00 5.00 4.05 0.59 0.35

C the coordination ability in emotional labor 4.00 1.00 5.00 4.00 0.60 0.37

D the reflective ability after emotional labor 4.00 1.00 5.00 3.99 0.57 0.33

E the application ability to emotional labor strategies 3.83 1.00 5.00 3.97 0.61 0.38

4.Provide complete final scale items

Recommendations:

Include the final 30-item scale in full (wording + subscales) as a table or appendix.

Justify all item deletions

①Include the final 30-item scale in full (wording + subscales) as a table or appendix According to the suggestion, I have explained the dimensions, question items, and corresponding scoring rules in the table, as shown in the following table:

Copy of the Emotional Labor Ability Scale for Kindergarten Teachers

dimension items Scoring principle

Emotional intelligence

I can judge a child's emotional state through their facial expressions

All items are scored using the Likert five point scoring method. The higher the score, the stronger the emotional labor ability of kindergarten teachers in the corresponding dimension

I can judge a child's emotional state by their tone of speech

I can judge a child's emotional state through their body movements

I can judge a child's emotional state through their physiological arousal (shortness of breath, tremors, etc.)

I can judge the compound emotions of young children (such as anger, restlessness, surprise, etc.) by observing their words and expressions

I can understand why young children feel happy

I can understand why young children feel scared

I can understand why young children feel angry

The Ability of Internalizing Emotional Labor Rrules

I am familiar with the emotional expression norms that kindergarten teachers need to follow

I strongly agree with the normative requirements that kindergarten teachers should follow when expressing emotions, such as gentleness, kindness, enthusiasm, etc

My attitude towards the normative requirements that kindergarten teachers should follow in expressing their emotions and behaviors is relatively stable

I firmly believe that the emotional expression behavior of kindergarten teachers must follow certain normative requirements

I strongly advocate that kindergarten teachers must follow certain normative requirements when expressing their emotions

The Application Ability To Emotional Labor Strategies When interacting with young children, if I need to show excitement, I can feel it from the bottom of my heart

When interacting with young children, if I need to show enthusiasm, I can feel it from the bottom of my heart

When interacting with young children, if I need to show a relaxed mood, I can feel it from the bottom of my heart

When interacting with young children, I can naturally show a sad emotional state to meet work needs (such as reaching empathy with young children)

When interacting with young children, I can naturally show an angry emotional state to meet work needs (such as expressing negation of problematic behavior in young children)

When interacting with young children, I can naturally show a tense emotional state to meet work needs (such as attracting children's attention, etc.)

The Coordination Ability In Emotional Labor

When dealing with emotional events, I can still ensure the smooth progress of children's morning activities

When dealing with emotional events, I can still ensure the smooth progress of collective teaching activities for young children

When dealing with emotional events, I can still ensure the smooth progress of children's daily activities (such as eating, drinking, washing, toileting, sleeping, leaving the kindergarten, etc.)

When dealing with emotional events, I can still ensure the smooth progress of activities in the children's activity area

When dealing with emotional events, I can still ensure the smooth progress of outdoor activities for young children

When dealing with emotional events, I can still ensure the smooth progress of children's activities outside the kindergarten

The Reflective Ability After Emotional Labor I can reflect on the rationality of my emotional expression behavior based on my experience of emotional interaction with young children

I can reflect on the rationality of my emotional expression behavior based on my own emotional expression experience

I can reflect on whether my emotional expression behavior contributes to the development of children's emotional intelligence based on relevant theoretical knowledge

I can reflect on whether my emotional expression behavior helps meet the emotional needs of young children based on relevant theoretical knowledge

I can reflect on whether my emotional expression behavior meets the emotional requirements of kindergarten regulations based on relevant theoretical knowledge

I can persistently reflect on whether my emotional expression and behavior are reasonable

When my mood is relatively low, I can still persist in reflecting on whether my emotional expression behavior is reasonable

To make my emotional expression more in line with expectations, I am able to actively reflect on my emotional expression behavior on a continuous basis

②Justify all item deletions

I have explained the rationality of deleting items in the corresponding section of the paper, as follows:

Using principal component analysis and maximum variance method to extract common factors, and then deleting items with factor load value less than 0.5, commonality less than 0.4, multiple loads with similar load values, or improper factor classification, a total of “E1, E2, E3, B2” items were deleted, with 33 items remaining.

---

## [Editor Report · Decision Letter 4]

Development of Emotional Labor Ability Scale for Kindergarten Teachers

PONE-D-24-16344R4

Dear Dr. Juan Hong,

We’re pleased to inform you that your manuscript has been judged scientifically suitable for publication and will be formally accepted for publication once it meets all outstanding technical requirements.

Kind regards,

Mohamed Ahmed Said, Ph.D.

Academic Editor

PLOS ONE
---

## [Editor Report · Acceptance letter]

PONE-D-24-16344R4

PLOS ONE

Dear Dr. Hong,

I'm pleased to inform you that your manuscript has been deemed suitable for publication in PLOS ONE. Congratulations! Your manuscript is now being handed over to our production team.

Kind regards,

on behalf of

Dr. Mohamed Ahmed Said

Academic Editor

PLOS ONE